# GUI Knowledge Bench: Revealing the Knowledge Gap Behind VLM Failures in GUI Tasks

## Abstract

Large vision–language models (VLMs) have advanced graphical user interface (GUI) task automation but still lag behind humans. We hypothesize this gap stems from missing core GUI knowledge, which existing training schemes (such as supervised fine-tuning and reinforcement learning) alone cannot fully address. By analyzing common failure patterns in GUI task execution, we distill GUI knowledge into three dimensions: (1) interface perception, knowledge about recognizing widgets and system states; (2) interaction prediction, knowledge about GUI interaction conventions; and (3) instruction understanding, knowledge about procedural knowledge of GUI operations. We further introduce GUI Knowledge Bench, a benchmark with multiple choice and yes/no questions across six platforms (Web, Android, MacOS, Windows, Linux, IOS) and 292 applications. Our evaluation shows that current VLMs identify widget functions but struggle with perceiving system states, predicting actions, and interpreting task goals. Experiments on real world GUI tasks further validate the close link between GUI knowledge and task success. By providing a structured framework for assessing GUI knowledge, our work supports the selection of VLMs with greater potential prior to downstream training and provides insights for building more capable GUI agents.

## 1 Introduction

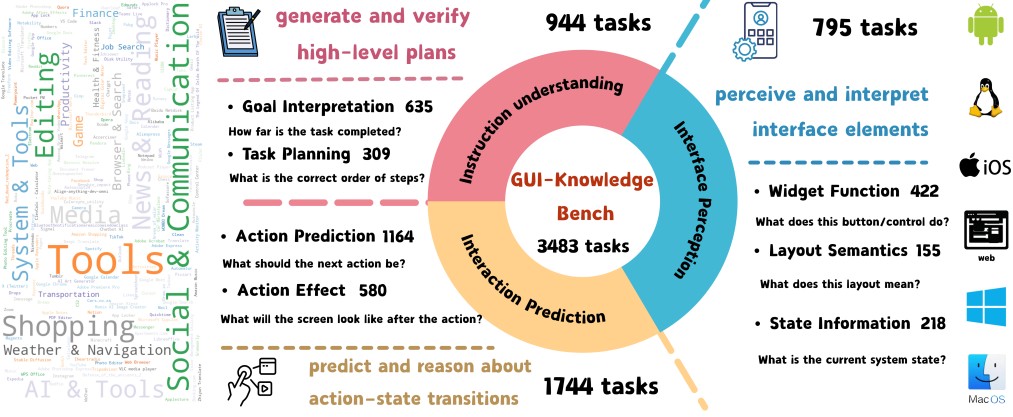

Figure 1: GUI Knowledge Bench: A benchmark evaluating VLMs on GUI knowledge across six platforms (Web, Android, MacOS, Windows, Linux, IOS). It measures three types of knowledge: Interface Perception, which evaluates understanding of GUI components, layout, and system state; Interaction Prediction, which assesses the knowledge of GUI interaction conventions; and Instruction Understanding, which tests whether a model knows the procedural knowledge of completing a GUI task.

Graphical User Interface (GUI) task automation, such as booking a flight, editing a presentation, or configuring system settings, poses unique challenges for AI agents (Wu et al., 2024a; Hong et al., 2024; Xu et al., 2024a; He et al., 2024). Recent approaches have leveraged large vision–language

models (VLMs) with techniques such as prompt engineering (Agashe et al., 2025; Xie et al., 2025a), supervised fine-tuning (SFT) (Wu et al., 2024b; Hong et al., 2024; Lin et al., 2025; Liu et al., 2025; Xu et al., 2024b), and reinforcement learning (RL) (Lian et al., 2025; Luo et al., 2025), achieving strong task performance in many applications. However, GUI agents still fail in many real-world scenarios (Xie et al., 2025c). For example, agents may misinterpret widget functions in unfamiliar applications, fail to predict correct action parameters, or struggle with multi-step planning and error recovery in long-horizon GUI tasks. Our analysis suggests that a primary reason for these failures is that the used VLMs lack the necessary GUI knowledge. While prompt engineering, SFT, and RL can improve reasoning, grounding, and planning abilities, they contribute little to injecting new GUI knowledge (Ovadia et al., 2024), which also plays important roles in solving GUI tasks.

Different from most existing benchmarks that primarily evaluate task success, which mainly focus on the grounding (Li et al., 2025; Cheng et al., 2024; Jurmu et al., 2008), reasoning, and planning (Lin et al., 2024) capabilities of GUI agents, our work targets the missing dimension of knowledge evaluation. To systematically examine these knowledge gaps, we introduce GUI-Knowledge Bench, a benchmark designed to assess the extent of GUI knowledge encoded in VLMs prior to downstream tasks, while also serving as a diagnostic tool to guide the design of VLM-based agent systems. The benchmark is constructed from over 40,000 screenshots and 400 execution trajectories spanning 292 applications across six platforms (Web, Android, MacOS, Windows, Linux, IOS)). Through a combination of automated generation and manual annotation, we derive a set of 3483 knowledge-centric questions that systematically test VLMs' knowledge in GUI.

We categorize the GUI knowledge into three complementary aspects derived from common agent failure modes: (1) interface perception, which involves recognizing widget functions, layout semantics, and perceiving state information (e.g., enabled/disabled, selected/focused); (2) interaction prediction, which involves assessing knowledge of GUI interaction conventions (e.g., what changes after toggling a switch or submitting a form, and which parameters are required); and (3) instruction understanding, which focuses on grounding natural-language instructions into executable, multi-step operation sequences with coherent plans. This categorization enables a systematic examination of which components of GUI knowledge are already present in current models and which remain underdeveloped.

Our evaluation reveals that current VLMs are still short of enough knowledge in these three categories for completing real world GUI tasks. First, although VLMs perform well at discerning different widget functions and layout semantics but struggle to accurately perceive system states. Second, VLMs underperform in interaction prediction, showing difficulties in anticipating correct action outcomes and required parameters. They frequently confuse click actions with other types of actions, a behavior commonly observed in many models. Third, VLMs struggle with judging task completion states and understanding human instructions. Some tasks are easy to complete, yet they still fail because the models do not understand the goals of the tasks. These findings highlight critical gaps in the internal GUI knowledge of current VLMs, revealing that while they can perceive interface elements, their understanding about system states and interaction outcomes remains limited. Our contributions are as followed:

- We introduce GUI-Knowledge bench, designed to evaluate GUI knowledge in both both general and GUI-tuned VLMs. Experiments on real world GUI environment further validates the close link between GUI knowledge and task success.
- Our evaluation identifies key gaps in perceiving system states, understanding the effect of common GUI interactions, and judging task completion, providing guidance for selecting or training VLMs prior to downstream GUI tasks.

## 2 RELATED WORK

### 2.1 GUI AGENT

Progress in GUI task automation has largely relied on pretrained vision–language models (VLMs), with improvements driven by supervised fine-tuning (SFT), reinforcement learning (RL), and synthetic data generation. SFT-based methods train VLMs on large-scale GUI datasets to enhance element grounding and action prediction, as seen in OS-Atlas (Wu et al., 2024b), CogAgent (Hong et al., 2024), and ShowUI (Lin et al., 2025), while multi-stage pipelines such as InfiGUIAgent (Liu

Table 1: Comparison of existing GUI benchmarks and our proposed benchmark across evaluation scope, operating system coverage, application diversity, and data scale. Our benchmark systematically spans multiple OS and applications with a comprehensive scope of GUI knowledge evaluation.

| Benchmark | Scope | OS | Apps | Task Num. |
|-----------|-------|-----|------|-----------|
| ScreenSpot-Pro (Li et al., 2025) | Action | 3 | 23 | 1581 |
| SeeClick (Cheng et al., 2024) | Action | 5 | 20+ | 1272 |
| VideoGUI (Lin et al., 2024) | Task | 1 | 11 | 463 |
| OSWorld (Xie et al., 2025c) | Task | 1 | 9 | 369 |
| MacOSworld (Yang et al., 2025) | Task | 1 | 30 | 202 |
| AndroidWorld (Rawles et al., 2024) | Task | 1 | 20 | 116 |
| MMBench-GUI (Wang et al., 2025) | Knowledge | 6 | - | 8000+ |
| Web-CogBench (Guo et al., 2025) | Knowledge | 1 | 14 | 876 |
| **GUI-Knowledge-Bench** | Knowledge | **6** | **292** | **3483** |

et al., 2025) and Aguvis (Xu et al., 2024b) further inject reasoning and planning abilities with synthetic data. RL approaches, including UI-AGILE (Lian et al., 2025) and GUI-R1 (Luo et al., 2025), refine action selection through long-horizon rewards or policy optimization, sometimes achieving superior performance with less training data. To address data scarcity, OS-Genesis () and UI-Genie (Sun et al., 2024) generate high-quality synthetic trajectories, while multi-agent systems such as GUI-OWL and Mobile-Agent-v3 (Wanyan et al., 2025) decompose perception, reasoning, and planning across modules to improve robustness in long-horizon tasks.

Despite these advances, most approaches primarily optimize execution strategies—whether through imitation of expert trajectories, reward shaping, or modular design—without fundamentally enriching the model's internal GUI knowledge. The trained models still fall short in interacting with unfamiliar applications or understanding complex system states. To address this gap, our work systematically evaluates these foundational knowledge deficiencies and introduces a benchmark that identifies missing GUI knowledge in VLMs prior to downstream training, providing insights into how future approaches may extend beyond standard fine-tuning paradigms.

## 2.2 GUI BENCHMARK

Evaluating GUI agents is essential for advancing their capabilities, and existing benchmarks generally fall into three categories. Action-level benchmarks focus on the precision of low-level operations such as mouse and keyboard inputs and accurate element grounding. Examples include ScreenSpot-Pro (Li et al., 2025) highlights grounding challenges in professional high-resolution interfaces, SeeClick (Cheng et al., 2024) and ScreenSpot (Jurmu et al., 2008) for cross-environment grounding. In contrast, we intentionally decouple grounding from the evaluation (by providing visual marks on screenshots), so that we can isolate and measure the knowledge deficits of current VLMs in GUI interactions, which other grounding-based benchmarks cannot reveal. Plan-level evaluations extend beyond single actions to hierarchical execution. VideoGUI (Lin et al., 2024), for instance, evaluates GUI agents with high-level and mid-level planning. Task-level benchmarks emphasize end-to-end task success in simulated environments, such as OSWorld (Xie et al., 2025c), OSWorld-Verified (Xie et al., 2025b), MacOSworld (Yang et al., 2025), and AndroidWorld (Rawles et al., 2024). Beyond execution, a few recent efforts assess GUI knowledge, such as MMBench-GUI (Wang et al., 2025), which tests content understanding and widget semantics, and Web-CogBench (Guo et al., 2025), which probes cognitive reasoning in web navigation. However, these benchmarks remain narrow in application scopes and domain knowledge coverage.

Our benchmark carefully categorizes the GUI knowledge into three complementary aspects derived from common agent failure modes, interface perception, interaction prediction and goal interpretation. Our benchmark offers a systematic and comprehensive evaluation of GUI knowledge, spanning multiple platforms and applications, thereby providing a more complete evaluation of base model's GUI knowledge.

## 3 GUI KNOWLEDGE BENCH

### 3.1 BENCHMARK OVERVIEW

We introduce GUI Knowledge Bench, a benchmark for systematically evaluating the knowledge VLMs need to complete GUI tasks. Based on common failure patterns in GUI task execution, we identify three complementary dimensions: interface perception, which covers recognizing GUI elements, their states, and layout semantics; interaction prediction, which tests whether models understand the effect and conventions of common GUI interactions; and instruction understanding, which examines whether models can interpret task goals and know the procedural knowledge of completing a GUI task. Together, these dimensions capture the core knowledge required for reliable GUI task completion and form the foundation of our benchmark.

### 3.2 DATA SOURCES AND COLLECTION PIPELINE

To build GUI Knowledge Bench, we aggregate data from multiple sources to ensure both trajectory-level interaction coverage and diverse standalone screenshots.

We leverage existing benchmarks such as GUI-Odyssey (Lu et al., 2024) and VideoGUI (Lin et al., 2024), which provide screenshots paired with tasks and action annotations. In addition, we collect new trajectories by running UI-Tars-7B agents in environments including OSWorld and MacOS-World, capturing realistic interaction sequences across both mobile and desktop platforms.

To further increase visual diversity and cover a wider range of application interfaces and operating systems, we further gather standalone GUI screenshots. Specifically, we sample from ScreenSpot v2 and extract representative key frames from YouTube tutorials, ensuring coverage of real-world applications, operating systems, and interface layouts. For less common actions, we manually perform operations on MacOS, Linux, and Windows, recording screenshots and corresponding actions.

Together, these sources yield a heterogeneous pool of GUI images and trajectories. From this pool, we construct task-specific question–answer pairs for each evaluation dimension, ensuring sufficient diversity and coverage while minimizing redundancy. Please refer the appendix for detailed statistics of our benchmark.

### 3.3 INTERFACE PERCEPTION

A fundamental requirement for completing GUI tasks is the ability to accurately perceive and interpret interactive elements in GUI. We aim to evaluate whether VLMs possess sufficient knowledge about graphical interfaces.

Specifically, this dimension encloses three aspects: (i) widget function understanding, i.e., recognizing the roles of common interface elements (e.g., three vertical dots for settings, speech bubbles for messaging apps); (ii) state information understanding, such as detecting whether a button is enabled/disabled, selected/focused, or toggled on/off; and (iii) layout semantics understanding, where spatial arrangement encodes critical information (e.g., distinguishing departure and arrival cities by their relative positions, identifying senders and receivers in an email, or inferring file hierarchy from indentation). Correctly perceiving these cues is essential for grounding subsequent reasoning and action.

**Task Definition.** We formalize the evaluation as a unified multiple-choice question-answering task. Given a question $q$, a set of candidate options $O$, and a screenshot $S$, the model is required to select the correct answer $o^*$ and provide its reasoning in thought $t$: VLM : $(S, q, O) \mapsto (t, o^*)$.

Our questions include two types: (1) multiple-choice with four candidates, and (2) judgment with Yes/No/Unknown. To reduce the burden of visual grounding, the relevant regions in the screenshot $S$ are highlighted using red dots or bounding boxes. This design ensures the evaluation focuses on whether the model possesses the required GUI knowledge rather than its grounding ability.

**Task Collection and Curation.** To construct the evaluation set, we first have human annotators design an initial set of seed questions based on the collected GUI screenshots. We then leverage GPT-5 to expand this pool with additional candidate questions, increasing diversity while maintaining relevance. Questions that can be answered based solely on the text, without viewing the screenshot, are

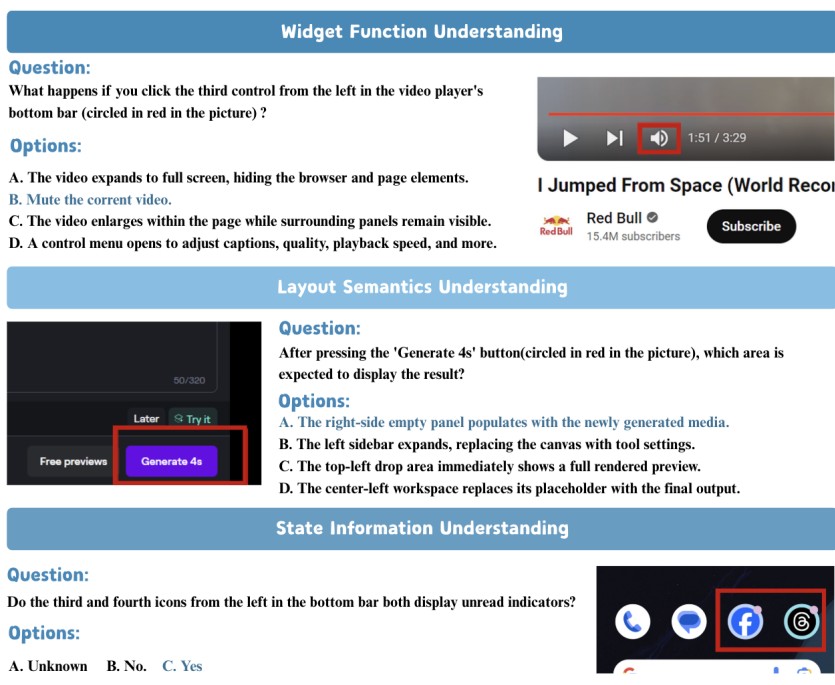

Figure 2: Example questions for Interface Perception. red bounding box

removed using Qwen-2.5-VL-7B to ensure visual understanding is necessary. Finally, the remaining questions are manually verified for correctness, and relevant regions in the screenshots are annotated to support precise visual grounding. This pipeline ensures that the evaluation focuses on interface perception knowledge rather than being confounded by grounding or annotation errors.

## 3.4 INTERACTION PREDICTION

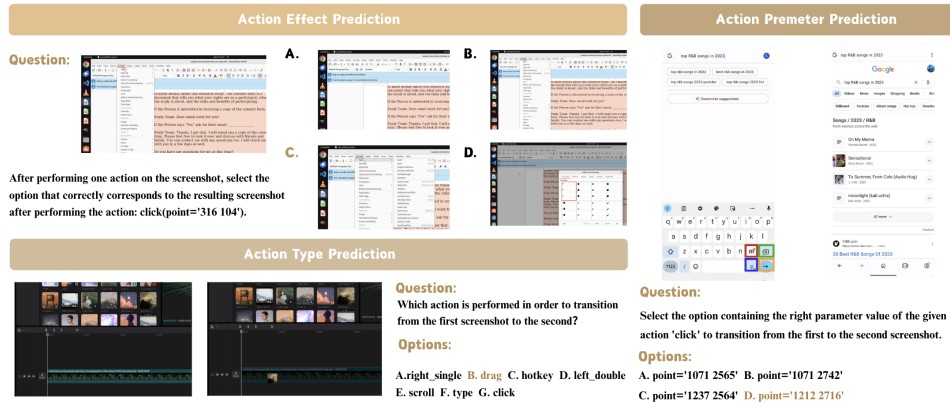

Figure 3: Example questions for Interaction Prediction.

A core requirement for solving GUI tasks is to know the interaction conventions and the effect of GUI operations. Unlike physical environments, GUI interactions follow symbolic and platform-specific rules (e.g., toggling a switch, typing text, dragging windows), which are often subtle and context-dependent. Without a proper understanding of these interaction conventions, models cannot reliably predict the consequences of actions or predict right action types/parameters to complete a GUI task. This motivates our evaluation of whether VLMs know GUI interaction conventions.

Interaction prediction is evaluated through two complementary tasks: (i) Action effect prediction, where the model is provided with a current screenshot $S$ and an action $a$, and must select the resulting screenshot $S'$ from a set of candidate options; (ii) Action prediction, where the model is given two consecutive screenshots $(s, s')$ and must infer the action $a$ that caused the transition. Action prediction is further divided into action type prediction, which identifies the correct action category, and action parameter prediction, which selects the appropriate arguments such as click coordinates, typed content, or drag vectors. We do not require the model to generate precise parameters of these coordinates. Exact coordinates in options are annotated in the image as shown in Figure 3. Therefore, the grounding demand is minimal.

**Task Definition.** We formalize GUI interaction dynamics as a state–action transition $S + a \to S'$, where $S$ represents the current screenshot, $S'$ the consequent screenshots and $a$ the action $a = (a_{\text{type}}, a_{\text{param}})$. (i) Action Effect Prediction. The model is given $S$ and $a$, and is required to select the resulting screenshots from a set of candidate screenshot options $O$: VLM : $(S, a, O) \mapsto S'$. (ii) Action Prediction. The model is given two consecutive screenshots $(S, S')$ and a set of candidate action types $O_{\text{type}}$, and must select the correct action type $a_{\text{type}}$: VLM : $(S, S', O_{\text{type}}) \mapsto a_{\text{type}}$. Given the correct action type $a_{\text{type}}$ and the same state pair $(S, S')$, the model selects the correct action parameters from a candidate set $O_{\text{param}}$: VLM : $(S, S', a_{\text{type}}, O_{\text{param}}) \mapsto a_{\text{param}}$.

**Task Collection and Curation.** To construct challenging distractor options, we design task-specific strategies for different prediction settings. For action effect prediction, candidate screenshots include the preceding screenshots, the true next screenshot, and visually similar but different screenshots sampled from current trajectories. For action type prediction, the model must choose from the full set of possible actions defined for the platform, with a unified action space across different platforms. (seven actions for desktop platforms and four actions for mobile platforms) For action parameter prediction, For clicks, bounding boxes of candidate elements are identified using OmniParser, and nearby but incorrect coordinates are sampled; for drags, distractors include reversed directions, shortened distances, or swapped start and end points; for scrolls, distractors vary in direction (up, down, left, right); for typing, inputs are perturbed with case changes, partial deletions, or common typos; and for hotkeys, distractors are drawn from a predefined set of common shortcuts. This design ensures that solving the tasks requires precise reasoning about GUI action dynamics rather than relying on superficial visual or layout cues.

## 3.5 INSTRUCTION UNDERSTANDING

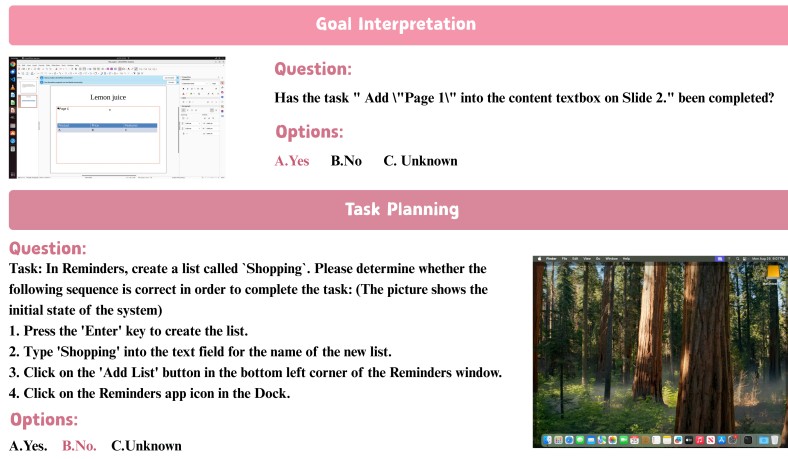

Figure 4: Example questions for Instruction Understanding.

Instruction understanding evaluates whether a model can interpret natural-language tasks and map them to a sequence of GUI operations. This ability is critical because many failures in GUI automation stem from misunderstanding task goals or misinterpreting user intent. Accurately understanding instructions and ordering a feasible sequence of steps is essential for performing high-level tasks in GUI environments. The questions do not ask the model to construct a plan or generate sub-steps.

Instead, the complete operation sequence is already provided in these questions to minimize the reasoning load.

We assess two complementary abilities: (i) goal interpretation, which evaluates whether a model can determine if a task has been successfully completed based on history screenshots; and (ii) task planning, which evaluates whether a model can reorder a set of candidate option steps into the correct sequence required to achieve the task goal. Together, these tasks test the model's ability to both verify and choose high-level plans. Once the underlying procedural knowledge is known, the answer becomes immediately obvious and requires minimal reasoning.

**Task Definition.** For goal interpretation, the model receives a natural-language task description $t$ and history screenshots, and must select the correct option $o^* \in \{\text{Yes}, \text{No}, \text{Unknown}\}$ indicating whether the task is completed: $\text{VLM} : (S_{1:T}, t, O) \mapsto o^*$. For task planning, the model is given a natural-language task description $t$, current screenshot $S$ and a set of candidate orderings $O = \{\pi_1, \pi_2, \ldots, \pi_m\}$, where each $\pi_i$ is a possible permutation of the operation steps. The model must select the correct ordering $\pi^*$ from $O$: $\text{VLM} : (t, S, O) \mapsto \pi^*$.

**Task Collection and Curation.** For goal interpretation, human annotators label each trajectory as successful or unsuccessful based on last one to five screenshots of the trajectory, and some successful trajectories are augmented by removing the final screenshot to create unsuccessful ones. For task planning, operation plans are first generated by Chat-GPT-5 and then verified by annotators. The annotated steps are automatically shuffled to form multiple-choice ordering questions, with longer sequences retaining initial steps and only permuting later steps. For shorter sequences, additional question formats are created by converting the shuffled sequence into Yes/No/Unknown questions, or into operation-level fill-in-the-blank questions with distractor steps. Tasks solvable without observing screenshots are filtered out using Qwen-VL-2.5-7B ensuring the difficulty of the questions.

# 4 BENCHMARKING VLMS

## 4.1 SETTINGS

We evaluate a diverse set of both open- and closed-source models on the GUI Knowledge Bench. The closed-source set includes Claude-Sonnet-4-5 (Anthropic, 2025a), Claude-Sonnet-4 (Anthropic, 2025b), Doubao-V-Pro (Doubao-1.5-Thinking-Vision-Pro-250428) (Team, 2025b), Gemini-2.5-Pro (Gemini Team, 2025), GPT-5-chat (OpenAI, 2025a), O3(OpenAI, 2025b), and GLM-4.5 (Team, 2025c). The open-source set covers Qwen2.5-72B (Qwen2.5-VL-72B-Instruct), Qwen2.5-7B (Qwen2.5-VL-7B-Instruct) (Team, 2025d), Qwen3-vl-8b-thinking Team (2025e). Besides we also include GUI finetune models such as UITARS-1.5-7B (Team, 2025a) and GUI-OWL-7b (Ye et al., 2025). Apart from necessary model-specific settings, all other parameters (e.g., temperature, top-p) were kept consistent across evaluations. Please refer the appendix for the detailed message template for each knowledge categories.

## 4.2 BENCHMARKING RESULTS

Table 2 summarizes the performance of all evaluated models on three knowledge categories. Experimental results highlight the following key observations.

**First**, o3 achieves strong performance across multiple metrics, consistent with its high success rate in real GUI tasks; notably, in the OSWorld benchmark under the Agent framework category, four of the top five agents leverage o3 (e.g., Agent-S2.5 w/ O3 50-step version and 100-step version, Jedi-7B w/ O3 w/ 50-step version and 100-step version). (Agashe et al., 2025; Xie et al., 2025a) This is likely because o3 effectively replaces the auxiliary modules that were removed or made optional.

**Second**, UITARS-1.5-7B, trained on Qwen2.5VL-7B, shows improvements in instruction understanding and goal reasoning but a decline in interface perception. Upon examining model outputs, we note the following recurring error types of UITARS-1.5-7B, thought–action mismatch where the internal reasoning conflicts with the executed code or selection; weakened instruction following that overlooks visual prompts and answers from prior knowledge; and degraded general perception, with frequent mistakes on basic attributes such as quantity, color, and length.

**Third**, smaller models retain only limited knowledge, suggesting that retrieval-augmented generation or knowledge-base integration may be a viable approach to enhance GUI agent performance.

Table 2: Performance on GUI Knowledge Bench across three dimensions. Bold numbers indicate the best results in each sub-task.

| Model | Interface Perception | | | Interaction Prediction | | | Instruction Understanding | | Overall |
|---|---|---|---|---|---|---|---|---|---|
| | state | widget | layout | effect | type | parameter | goal | plan | |
| O3 (OpenAI, 2025b) | **83.03%** | 84.12% | **88.39%** | **74.83%** | 75.98% | 45.75% | 69.45% | **95.47%** | **73.30%** |
| Gemini-2.5-Pro (Gemini Team, 2025) | 81.19% | 84.36% | 87.10% | 71.03% | 73.25% | **46.97%** | 67.72% | 92.56% | 71.69% |
| GPT-5-Chat (OpenAI, 2025a) | 78.90% | 84.12% | **88.39%** | 71.55% | 71.55% | 43.85% | 68.98% | 91.26% | 70.97% |
| Qwen3-vl-8b-thinking (Team, 2025e) | 68.81% | 76.30% | 83.23% | 67.07% | 70.36% | 40.73% | 64.09% | 91.26% | 66.81% |
| Claude-Sonnet-4-5 (Anthropic, 2025a) | 74.77% | 81.52% | 82.58% | 49.83% | 70.19% | 43.33% | **70.30%** | 91.56% | 66.53% |
| Qwen2.5VL-72B (Team, 2025d) | 69.27% | 77.49% | 80.00% | 61.72% | 64.91% | 38.99% | 62.20% | 85.44% | 63.88% |
| Doubao-V-Pro (Team, 2025b) | 72.48% | 83.65% | 81.29% | 67.24% | 75.64% | 41.07% | 33.07% | 94.17% | 63.42% |
| Claude-Sonnet-4 (Anthropic, 2025b) | 70.18% | 78.44% | 78.06% | 41.90% | 62.52% | 42.11% | 65.20% | 94.82% | 62.16% |
| Qwen2.5VL-7B (Team, 2025d) | 53.21% | 67.77% | 60.00% | 51.72% | 50.60% | 39.34% | 16.22% | 48.87% | 45.16% |
| UITARS-1.5-7B (Team, 2025a) | 49.54% | 59.48% | 59.35% | 22.24% | 59.11% | 34.32% | 38.74% | 55.34% | 44.27% |
| GUI-OWL-7b (Ye et al., 2025) | 60.09% | 64.93% | 63.23% | 21.55% | 55.37% | 36.05% | 21.26% | 39.81% | 40.74% |
| GLM-4.5 (Team, 2025c) | 49.54% | 48.10% | 53.55% | 27.07% | 17.55% | 35.53% | 28.98% | 91.91% | 38.10% |

## 4.3 ERROR ANALYSIS AND DISCUSSION

### 4.3.1 INTERFACE PERCEPTION.

Most models handle widget functions and layout semantics well but struggle with system state perception. As shown in Figure 5, in Safari, an update notification is often mistaken for a blocking pop-up, leading to incorrect predictions, while in PowerPoint, models can understand the effect of delete action but not which element is selected. Our benchmark reveals that current models underperform in system state perception, despite its crucial role in GUI tasks.

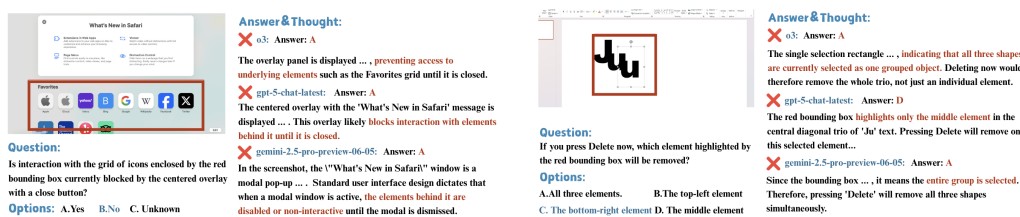

Figure 5: Example failure cases of interface perception questions.

### 4.3.2 INTERACTION PREDICTION.

On desktop, models often confuse click, double-click, and right-click. This is partly because different operating systems and applications treat these interactions differently: in some contexts, single, double, or right clicks can substitute for each other, while in others the distinction is strict. Humans often try multiple actions to achieve a goal, but models predict a single action based on learned patterns. As a result, less frequent actions like double-click or right-click are more prone to misprediction, especially for smaller models. Please refer to appendix to see the confusion matrix of action type prediction.

### 4.3.3 INSTRUCTION UNDERSTANDING.

Our benchmark highlights a failure in goal understanding. In Figure 6, three models (o3, gpt-5-chat-latest, and gemini-2.5-pro) all respond "Yes," claiming that the Freeform icon is gone from the Dock, even though a different app was removed. This shows that the models cannot reliably judge whether the requested task is actually completed.

## 4.4 RESULTS ON REAL-WORLD GUI TASKS

This section examines the role of three types of knowledge in enabling successful real-world GUI task execution.

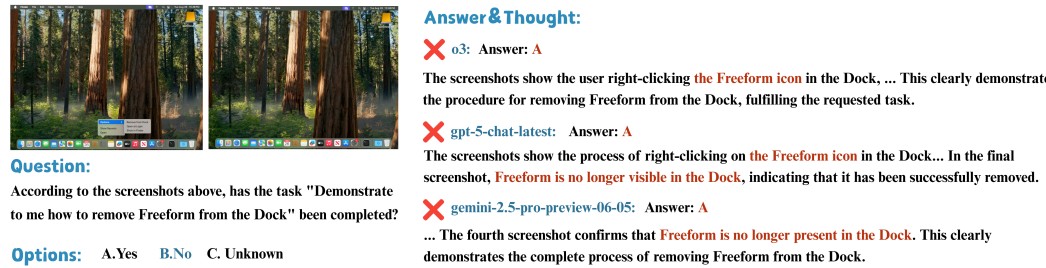

Figure 6: Example failure cases of instruction understanding questions.

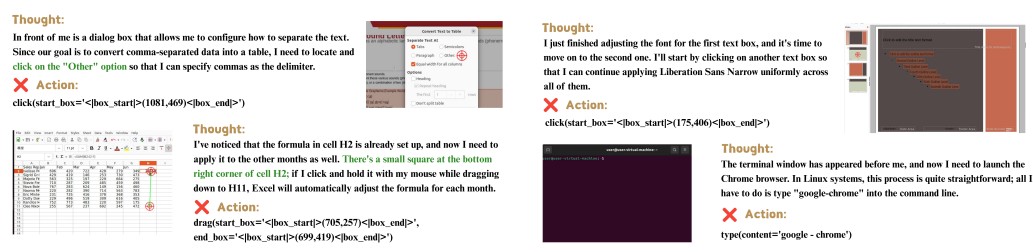

Figure 7: Example failure cases due to lack of interaction prediction knowledge.

### 4.4.1 QUALITATIVE ANALYSIS: INTERFACE PERCEPTION AND INTERACTION PREDICTION

**Interface Perception Knowledge.** In our evaluation, we identified several tasks that the model consistently failed to solve even under the pass@32 setting. We attribute some of them to lack of knowledge of the interface. Two example failure cases are shown in Figure 8. When asked to add a note, the model repeatedly attempted to insert comments or text boxes, incorrectly treating these actions as equivalent to adding a note. In reality, adding a note requires first enabling the Notes pane through the View menu and then placing the note at the bottom of the slide. Similarly, in converting comma-separated text into a table, the model repeatedly failed because it did not specify the delimiter, a necessary step for correct execution. These cases suggest that the failures stem from missing application-specific knowledge rather than inherent reasoning limitations. Importantly, once the required knowledge was provided, the model was able to complete these tasks successfully.

Table 3: Effect of appending operation plans on UITARS-1.5-7B.

| Metric | UITARS-1.5-7B (base) | + GPT-4o plan | + OSWorld-human plan | + o3 plan |
|---|---|---|---|---|
| Pass@1 (%) | 24.81 | 27.59 | 28.20 | **30.79** |

**Interaction Prediction Knowledge.** As shown in Figure 7, many errors of models occur because it lacks knowledge for localizing interface elements correctly. Prior work has improved localization using masks, accessibility trees, or APIs. Another promising approach is to leverage actions themselves for self-verification, using visual prompts to check if the executed action was correct.

### 4.4.2 QUANTITATIVE ANALYSIS: THE IMPACT OF PLAN INJECTION

**Instruction Understanding Knowledge.** We generated knowledge about operation plans from GPT-4o and o3 conditioned on task instructions, and used human-authored operation plans from OSWorld-human. Each plan was appended to the original task description as an additional knowledge for UITARS-1.5-7B. Results are summarized in Table 3. These results show that providing knowledge about operation plans improves task performance, highlighting the importance of instruction understanding for task completion. Notably, o3-generated plans achieve the largest gain, surpassing human-authored plans and aligning with o3's top performance across our benchmark evaluations.

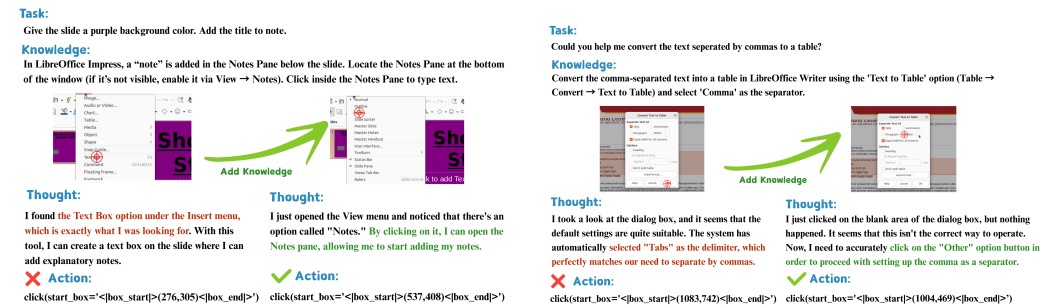

Figure 8: Example failure cases due to lack of interface perception knowledge.

### 4.4.3 VALIDATION STUDY: KNOWLEDGE AS A NECESSARY CONDITION

While the previous experiment demonstrates the benefit of injecting Instruction knowledge, Interface and Interaction knowledge are intrinsic and difficult to inject externally. To address this, we conduct a validation study that mirrors the the 'Instruction Understanding' analysis for 'Interface Perception' and 'Interaction Prediction'. Concretely, we transform 39 questions in our benchmark into practical GUI tasks, where the knowledge tested in questions is key to completing the GUI tasks. Two examples are as follows. From **Knowledge Question A**: "Are the search results limited to a specific region? (Yes/No/Unknown)" to **Transformed Task A**: "Set the search results restricted to a specific region: Japan." (Ground Truth: Click the toggle switch). From **Knowledge Question B**: "Is the list currently arranged from older to newer items?" to **Transformed Task B**: "Sort the items from oldest to newest according to the time added." (Ground Truth: Click the column header).

We define $S1$ as answering the knowledge question correctly, and $S2$ as successfully completing the corresponding GUI tasks. Results are shown in Table 4. Prompts for completing GUI tasks are provided in the Appendix.

Table 4: Correlation between GUI Knowledge ($S_1$) and Task Completion ($S_2$).

| Model | $P(S_2\checkmark|S_1\checkmark)$ | $P(S_2\times|S_1\checkmark)$ | $P(S_2\times|S_1\times)$ | $P(S_2\checkmark|S_1\times)$ |
|---|---|---|---|---|
| claude-sonnet-4 (Anthropic, 2025b) | 20.00% | 80.00% | 100.00% | 0.00% |
| claude-sonnet-4-5 (Anthropic, 2025a) | 8.33% | 91.67% | 100.00% | 0.00% |
| Doubao-V-Pro (Team, 2025b) | 0.00% | 100.00% | 100.00% | 0.00% |
| Gemini-2.5-Pro (Gemini Team, 2025) | 0.00% | 100.00% | 100.00% | 0.00% |
| GPT-5-Chat (OpenAI, 2025a) | 5.56% | 94.44% | 100.00% | 0.00% |

The results establish GUI knowledge as a necessary but not sufficient condition for task completion: Lacking knowledge guarantees execution failure (e.g., 100% for Gemini-1.5), validating that knowledge is the strict lower bound for control. Even with correct understanding, execution often fails due to grounding precision. This confirms that our benchmark measures a foundational capability. While having knowledge doesn't guarantee success (due to downstream grounding issues), lacking knowledge guarantees failure.

## 5 CONCLUSION

We introduces GUI Knowledge Bench, a novel benchmark designed to evaluate the GUI knowledge encoded in vision-language models (VLMs) before downstream training. By analyzing common failure patterns in GUI task execution, the benchmark categorizes GUI knowledge into three dimensions: interface perception, interaction prediction, and instruction understanding. The evaluation reveals significant gaps in current VLMs' understanding of system states, action outcomes, and task completion verification. These findings highlight the necessity of enriching VLMs with domain-specific GUI knowledge to enhance their performance in real-world GUI tasks and provide insights to guide the development of more capable GUI agents.

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

## A  APPENDIX

### A.1  QUESTION GENERATION PROMPT TEMPLATE FOR INTERFACE PERCEPTION

**Prompt for widget function understanding.**

---

**Widget Function Prompt**

**System Prompt:**

**[Role]**
You will be provided with a single screenshot of a system interface (desktop app, web UI, or mobile app). Generate exactly one challenging GUI reasoning question about that screenshot that requires inspecting the image to answer.

**[Knowledge Scope of the question]**
Ask about the intended function of a specific UI widget (button, toggle, slider, icon, etc.) inferred from the widget's iconography and surrounding context. Avoid universally trivial icons unless combined with contextual clues.

**[Generation Guidelines]**

1. Question length: one concise sentence only. No hints, no steps, no extra context.

2. Position-only references: Do NOT use any visible text, icon names, or labels from the screenshot. Refer ONLY by position or coordinates (examples: "top-right corner", "third from left in the top toolbar", "second row, third column", "left sidebar, bottom icon", or "`<x,y>`" with origin top-left). The question must be unsolvable without the screenshot.

3. Question types and options:
   - If `multiple_choice`: produce exactly 4 options. The first option MUST be the correct answer.
   - If `yes_or_no`: produce exactly 3 options: {"yes", "no", "unknown"} and the correct one must be first.
   - If the correct answer is genuinely not deducible from the screenshot or you cannot answer the correct answer, then use:
     – multiple_choice: first option = "none of the other options are correct."
     – yes_or_no: first option = "unknown"

4. Option style: Options must describe actions or effects (not icon shapes). Keep options parallel in length and style ($\approx$ 6–16 words).

5. Distractors: The 3 incorrect options must be plausible and similar to the correct one.

6. Contextual reasoning: Prefer questions requiring reasoning across UI elements (e.g., highlighted rows, active tab, enabled/disabled states, adjacent panels).

7. Based on the provided screenshot, identify which application is currently being used and include this information in your output JSON under the field `app_type`.

---

**[Output JSON schema — return exactly this JSON object (no extra text)]**

```
{
  "question_type": "multiple_choice" or "yes_or_no",
  "question_text": "<one concise sentence using only positions>",
  "option_text": ["<first option correct>", "<distractor 1>",
  ↪  "<distractor 2>", "<distractor 3>"],
  "app_type": "<application type of the current screenshot>",
  "os_type": "Linux" | "Windows" | "Android" | "MacOS" | "IOS" | "Web"
}
```

**[Example Output]**

```
{
  "question_type": "yes_or_no",
  "question_text": "While cell B5 in the 'First Name' column shows
  ↪  'Walter' in the formula bar and the checkmark and 'X' icons are
  ↪  visible beside it, will clicking the 'X' icon clear formatting in
  ↪  the selected cell",
  "option_text": ["yes","no","unknown"],
  "app_type": "Excel",
  "os_type": "Linux"
}
{
  "question_type": "multiple_choice",
  "question_text": "Which of the following statement is correct
  ↪  according to the screenshots?",
  "option_text": [
    "The camera is not currently connected to WiFi",
    "The camera can not be controlled remotely from the phone",
    "Pressing the 'phone' mode icon in the top bar can lead to turning
    ↪  on the phone's airplane mode",
    "Pressing the 'clone' mode icon in the top bar can lead to signing
    ↪  out of the cloud gallery"
  ],
  "app_type": "Excel",
  "os_type": "Linux"
}
```

**Prompt for layout semantics understanding.**

Layout Semantics Prompt

**System Prompt:**

**[Role]**
You will be provided with a single screenshot of a system interface (desktop app, web UI, or mobile app). Generate exactly one challenging GUI reasoning question about that screenshot that requires inspecting the image to answer.

**[Knowledge Scope of the question]**
The questions should assess whether the model understands positional and grouping relationships between UI elements, inferring their roles from placement and hierarchy.

**[Generation Guidelines]**

1. Question length: one concise sentence only. No hints, no steps, no extra context.

2. Position-only references: Do NOT use any visible text, icon names, or labels from the screenshot. Refer ONLY by position or coordinates (examples: "top-right corner", "third from left in the top toolbar", "second row, third column", "left sidebar, bottom icon", or "<x,y>" with origin top-left). The question must be unsolvable without the screenshot.

3. Question types and options:

   - If `multiple_choice`: produce exactly 4 options. The first option MUST be the correct answer.

- If `yes_or_no`: produce exactly 3 options: {"yes", "no", "unknown"} and the correct one must be first.
- If the correct answer is genuinely not deducible from the screenshot or you cannot answer the correct answer, then use:
  - multiple_choice: first option = "none of the other options are correct."
  - yes_or_no: first option = "unknown"

4. Option style: Options must describe actions or effects (not icon shapes). Keep options parallel in length and style ($\approx$ 6–16 words).

5. Distractors: The 3 incorrect options must be plausible and similar to the correct one.

6. Contextual reasoning: Prefer questions requiring reasoning across UI elements (e.g., highlighted rows, active tab, enabled/disabled states, adjacent panels).

7. Based on the provided screenshot, identify which application is currently being used and include this information in your output JSON under the field `app_type`.

**[Output JSON schema — return exactly this JSON object (no extra text)]**

```
{
  "question_type": "multiple_choice" or "yes_or_no",
  "question_text": "<one concise sentence using only positions>",
  "option_text": ["<first option correct>", "<distractor 1>",
  ↪   "<distractor 2>", "<distractor 3>"],
  "app_type": "<application type of the current screenshot>",
  "os_type": "Linux" | "Windows" | "Android" | "MacOS" | "IOS" | "Web"
}
```

**[Example Output]**

```
{
  "question_type": "multiple_choice",
  "question_text": "What is likely to be the departure city?",
  "option_text": ["Beijing", "Shanghai", "Guangzhou", "None of the
  ↪   other options."],
  "app_type": "website",
  "os_type": "Windows"
}

{
  "question_type": "yes_or_no",
  "question_text": "Is the folder in the second row under the
  ↪   'Documents' folder?",
  "option_text": ["yes", "no", "unknown"],
  "app_type": "Thunderbird",
  "os_type": "Windows"
}

{
  "question_type": "multiple_choice",
  "question_text": "Who sends this email. Please answer the email
  ↪   address.",
  "option_text": ["li@gmail.com", "zhang@gmail.com", "wang@gmail.com",
  ↪   "None of the other options."],
  "app_type": "Email",
  "os_type": "Windows"
}
```

**Prompt for state information understanding.**

State Information Prompt

**System Prompt:**

**[Role]**
You will be provided with a single screenshot of a system interface (desktop app, web UI, or mobile app). Generate exactly one challenging GUI reasoning question about that screenshot that requires inspecting the image to answer.

**[Knowledge Scope of the question]**
Ask about the current state information of the system, such as whether a control is enabled/disabled, a process is in-progress/completed, a request is pending, or the system is online/offline. Prefer reasoning that requires subtle visual cues or multi-element context.

**[Generation Guidelines]**

1. Question length: one concise sentence only. No hints, no steps, no extra context.

2. Position-only references: Do NOT use any visible text, icon names, or labels from the screenshot. Refer ONLY by position or coordinates (examples: "top-right corner", "third from left in the top toolbar", "second row, third column", "left sidebar, bottom icon", or "`<x,y>`" with origin top-left). The question must be unsolvable without the screenshot.

3. Question types and options:

    - If `multiple_choice`: produce exactly 4 options. The first option MUST be the correct answer.
    - If `yes_or_no`: produce exactly 3 options: {"yes", "no", "unknown"} and the correct one must be first.
    - If the correct answer is genuinely not deducible from the screenshot or you cannot answer the correct answer, then use:
        - multiple_choice: first option = "none of the other options are correct."
        - yes_or_no: first option = "unknown"

4. Option style: Options must describe actions or effects (not icon shapes). Keep options parallel in length and style ($\approx$ 6–16 words).

5. Distractors: The 3 incorrect options must be plausible and similar to the correct one.

6. Contextual reasoning: Prefer questions requiring reasoning across UI elements (e.g., highlighted rows, active tab, enabled/disabled states, adjacent panels).

7. Based on the provided screenshot, identify which application is currently being used and include this information in your output JSON under the field `app_type`.

**[Output JSON schema — return exactly this JSON object (no extra text)]**

```
{
  "question_type": "multiple_choice" or "yes_or_no",
  "question_text": "<one concise sentence using only positions>",
  "option_text": ["<first option correct>", "<distractor 1>",
  ↪  "<distractor 2>", "<distractor 3>"],
  "app_type": "<application type of the current screenshot>",
  "os_type": "Linux" | "Windows" | "Android" | "MacOS" | "IOS" | "Web"
}
```

**[Example Output]**

```
{
  "question_type": "multiple_choice",
  "question_text": "The button in the lower toolbar is active, but the
  ↪  button next to it is greyed out. Which condition is most likely
  ↪  not met yet?",
  "option_text": [
    "All required fields are filled",
    "Network connection is active",
    "File format is supported",
    "None of the other options"
  ],
  "app_type": "Form Editor",
  "os_type": "Web"
}
```

```
{
  "question_type": "multiple_choice",
  "question_text": "How can the user enable more controls over the
  ↪  alignment of objects?",
  "option_text": [
    "Select more than one object",
    "Double click the alignment button",
    "None of the other options",
    "User is logged in"
  ],
  "app_type": "Graphics Editor",
  "os_type": "Windows"
}

{
  "question_type": "yes_or_no",
  "question_text": "Will the option in the toolbar become available
  ↪  immediately after selecting a file?",
  "option_text": ["yes","no","unknown"],
  "app_type": "Document Editor",
  "os_type": "MacOS"
}

{
  "question_type": "yes_or_no",
  "question_text": "Is the movie export function currently available?",
  "option_text": ["no","yes","unknown"],
  "app_type": "Video Editor",
  "os_type": "Linux"
}
```

## A.2 PLAN GENERATION PROMPT TEMPLATE FOR OSWORLD TASKS.

---
**User Instruction Prompt**

**User Prompt:**

Analyze the given GUI task and break it down into essential, actionable steps. You will receive:
- a task instruction: `{task_instruction}` - the app where the task occurs: `{app_name}` -
the initial screenshot image
Your goal is to output a Python list of clear, concise steps in logical order to complete the task
within the app. Each step should represent a key state, action, or milestone. Use simple, direct
language. Avoid ambiguity or unnecessary complexity.
**Output format:**

- A valid Python list of strings, e.g.:

    `["First step.", "Second step.", "Third step."]`

- Each string must use double quotes ("), and the output must be directly parsable using
  `eval()` or `ast.literal_eval()`.

- Output only the list. No explanation, no extra text.

**Constraints:**

- Ensure each step is actionable and unambiguous,

- Ensure each step is necessary for task completion,

- Ensure each step is easy to follow by a user.

---

### A.3 EVALUATION MESSAGE PROMPT TEMPLATE

#### A.3.1 INTERFACE PERCEPTION.

All evaluation questions in this knowledge category use the same prompt template as shown below.

---

**GUI Agent Inference Prompt**

**System**
You are a Graphical User Interface (GUI) agent. You will be given a screenshot, a question, and corresponding options. You need to choose one option as your answer.

**User**
```
{question_images}
{question_texts}
{question_options}
```

**Response Rules**

**If question_type == 'yes_or_no':**
Think step by step. You must respond strictly in JSON format following this schema:

```
{
  "thought": "<your reasoning>",
  "answer": "<yes/no/unknown>"
}
```

**If question_type == 'multiple_choice':**
Think step by step. You must respond strictly in JSON format following this schema:

```
{
  "thought": "<your reasoning>",
  "answer": "<A/B/C/D>"
}
```

---

**Interaction Prediction.**

---

**GUI Agent Task-Solving Prompt**

**System**
You are a Graphical User Interface (GUI) agent. You will be given a task instruction, a screenshot, several GUI operations, and four options. Your goal is to select the best option that could solve the task.
```
{question_images}
```

**User**
```
{question_text}
```
Which of the above options are correct according to the screenshots? Think step by step. You must respond strictly in JSON format following this schema.

**Response Schema**

```
{
  "thought": "<your reasoning>",
  "answer": "<A/B/C/D>"
}
```

---

#### A.3.2 INTERACTION PREDICTION

**ActionEffect**

---

**GUI Agent Next-State Selection Prompt**

**System**

You are a Graphical User Interface (GUI) agent. You will be given a screenshot, action descriptions, and multiple options, each containing an image. After performing one action on the screenshot, your goal is to select the option that correctly corresponds to the resulting screenshot after performing the action. Below is a short description of the action space:

```
if platform == Desktop:
        Action Space
        - click(point='x1 y1'): left click a position on the screen.
        - left_double(point='x1 y1'): left double click a position on
        ↪   the screen.
        - right_single(point='x1 y1'): right single click a position on
        ↪   the screen.
        - drag(start_point='x1 y1', end_point='x2 y2'): drag the mouse
        ↪   from one position to another.
        - hotkey(key='ctrl c'): keyboard shortcut, split keys with
        ↪   spaces
        - type(content='xxx'): type an answer, use escape characters
        ↪   (', ", \n) when needed. Add \n at the end if it is the
        ↪   final submission.
        - scroll(point='x1 y1', direction='down or up or right or
        ↪   left'): scroll to see more content

if platform == Mobile:
        Action Space
        - click(point='x1 y1')
        - long_press(point='x1 y1')
        - type(content='') #If you want to submit your input, use "\\n"
        ↪   at the end of `content`.
        - scroll(point='x1 y1', direction='down or up or right or
        ↪   left'): scroll to see more content
```

The size of the image is {w}x{h}. \n

**User**

`{question_image}`
Above is the current screenshot.
After I perform the described action 'action_type(action_parameter)' (as drawn in the initial screenshot), which of the following options correctly corresponds to the resulting screenshot?
A. `{option_image_A}`
B. `{option_image_B}`
C. `{option_image_C}`
D. `{option_image_D}`

**Response Schema**

Think step by step. You must respond strictly in JSON format following this schema:

```
{
  "thought": "<your reasoning>",
  "answer": "<A/B/C/D>"
}
```

**ActionPrediction - Parameter**

---

**GUI Agent Action-Parameter Selection Prompt**

**System**

You are a Graphical User Interface (GUI) agent. You will be given two consecutive screenshots of the GUI, action descriptions, and multiple options. Your goal is to select which action was performed to transition from the first screenshot to the second. If the description specifies an action type, select the correct parameter value for the given action.

---

```
if platform == Desktop:
        Action Space
        - click(point='x1 y1'): left click a position on the screen.
        - left_double(point='x1 y1'): left double click a position on
        ↪   the screen.
        - right_single(point='x1 y1'): right single click a position on
        ↪   the screen.
        - drag(start_point='x1 y1', end_point='x2 y2'): drag the mouse
        ↪   from one position to another.
        - hotkey(key='ctrl c'): keyboard shortcut, split keys with
        ↪   spaces
        - type(content='xxx'): type an answer, use escape characters
        ↪   (', ", \n) when needed. Add \n at the end if it is the
        ↪   final submission.
        - scroll(point='x1 y1', direction='down or up or right or
        ↪   left'): scroll to see more content

if platform == Mobile:
        Action Space
        - click(point='x1 y1')
        - long_press(point='x1 y1')
        - type(content='') #If you want to submit your input, use "\\n"
        ↪   at the end of `content`.
        - scroll(point='x1 y1', direction='down or up or right or
        ↪   left'): scroll to see more content
```

The size of the image is {w}x{h}. \n
{question_images}

**User**

Above are two consecutive screenshots. Your task is to select the option containing the right parameter value of the given action ' {action_type} ' to transition from the first to the second screenshot.
As is drawn in the first screenshot. Which of the above options are correct according to the screenshots?
A. {option_text}
B. {option_text}
C. {option_text}
D. {option_text}

**Response Schema**
Think step by step. You must respond strictly in JSON format following this schema:

```
{
  "thought": "<your reasoning>",
  "answer": "<A/B/C/D>"
}
```

**ActionPrediction - Type**

---

**GUI Agent Action Identification Prompt**

**System**
You are a Graphical User Interface (GUI) agent. You will be given two consecutive screenshots of the GUI, action descriptions, and multiple options. Your goal is to select which action was performed to transition from the first screenshot to the second. If the description specifies an action type, select the correct parameter value for the given action.

```
if platform == Desktop:
        Action Space
        - click(point='x1 y1'): left click a position on the screen.
        - left_double(point='x1 y1'): left double click a position on
        ↪   the screen.
```

```
        - right_single(point='x1 y1'): right single click a position on
        ↪   the screen.
        - drag(start_point='x1 y1', end_point='x2 y2'): drag the mouse
        ↪   from one position to another.
        - hotkey(key='ctrl c'): keyboard shortcut, split keys with
        ↪   spaces
        - type(content='xxx'): type an answer, use escape characters
        ↪   (', ", \n) when needed. Add \n at the end if it is the
        ↪   final submission.
        - scroll(point='x1 y1', direction='down or up or right or
        ↪   left'): scroll to see more content

if platform == Mobile:
        Action Space
        - click(point='x1 y1')
        - long_press(point='x1 y1')
        - type(content='') #If you want to submit your input, use "\\n"
        ↪   at the end of `content`.
        - scroll(point='x1 y1', direction='down or up or right or
        ↪   left'): scroll to see more content
```

The size of the image is {w}x{h}. \n
`{question_images}`

**User**

Above are two consecutive screenshots. Your task is to select which action is performed in order to transition from the first screenshot to the second.

```
if platform == Desktop:
    {seven action types}
    Which of the above options are correct according to the
    ↪   screenshots?
    Think step by step. You must respond strictly in JSON format
    ↪   following this schema:
    {"thought": "<your reasoning>", "answer": "<A/B/C/D/E/F/G>" }

if platform == Mobile:
    {four action types}
    Which of the above options are correct according to the
    ↪   screenshots?
    Think step by step. You must respond strictly in JSON format
    ↪   following this schema:
    {"thought": "<your reasoning>", "answer": "<A/B/C/D>" }
```

**Response Schema (Desktop)**

```
{
  "thought": "<your reasoning>",
  "answer": "<A/B/C/D/E/F/G>"
}
```

**Response Schema (Mobile)**

```
{
  "thought": "<your reasoning>",
  "answer": "<A/B/C/D>"
}
```

### A.3.3 INSTRUCTIONUNDERSTANDING

**GoalInterpretation**

**Task Completion Verification Prompt**

**System**

You are a Graphical User Interface (GUI) agent. You will be given a sequence of screenshots, a task instruction, and three possible answer options: `yes`, `no`, `unknown`. Your goal is to select the best option that indicates whether the task is completed.

- **yes** — The task is clearly completed.

- **no** — The task is not completed.

- **unknown** — The screenshots do not provide enough evidence to determine completion.

**User**

According to the screenshots below, has the task "`{task}`" been completed?
`{question_images}`

**Response Schema**

Think step by step. You must respond strictly in JSON format following this schema:

```
{
  "thought": "<your reasoning>",
  "answer": "<yes/no/unknown>"
}
```

**TaskPlanning**

**GUI Agent Conditional QA Prompt**

**System**

**If `question_type == 'yes_or_no'`:**
You are a Graphical User Interface (GUI) agent. You will be given a screenshot, a question, and corresponding options. You need to choose one option as your answer.

**If `question_type == 'multiple_choice'`:**
You are a Graphical User Interface (GUI) agent. You will be given a task instruction, a screenshot, several GUI operations, and four options. Your goal is to select the best option that could solve the task.

`{question_images}`

**User**
`{question_text}`
`{option_texts}`
Which of the above options are correct according to the screenshot?

**Response Rules**

**If `question_type == 'yes_or_no'`:**
Think step by step. You must respond strictly in JSON format following this schema:

```
{
  "thought": "<your reasoning>",
  "answer": "<yes/no/unknown>"
}
```

**If `question_type == 'multiple_choice'`:**
Think step by step. You must respond strictly in JSON format following this schema:

```
{
  "thought": "<your reasoning>",
  "answer": "<A/B/C/D>"
}
```

## A.4 PROMPTS FOR COMPLETING GUI TASKS

---

**GUI Agent Conditional QA Prompt**

**System**

You are a helpful assistant.
**User**

You are a GUI agent. You are given a task and your action history, with screenshots. You need to perform the next action to complete the task.

**Output Format**
"' Thought: ... Action: ... "'

**Action Space**
click(start_box='<|box_start|>(x1,y1)<|box_end|>')
left_double(start_box='<|box_start|>(x1,y1)<|box_end|>')
right_single(start_box='<|box_start|>(x1,y1)<|box_end|>')
drag(start_box='<|box_start|>(x1,y1)<|box_end|>',
end_box='<|box_start|>(x3,y3)<|box_end|>')
hotkey(key='')
type(content='') If you want to submit your input, use "
n" at the end of 'content'.
scroll(start_box='<|box_start|>(x1,y1)<|box_end|>', direction='down or up or right or left')
done() If you think the instruction is finished, parameters none

**User Instruction**
Task Instruction

**Image Info**
Image size (pixels): width=image_size[0], height=image_size[1]. Output absolute pixel coordinates.

---

## A.5 DATASET STATISTICS OVERVIEW

We show more detailed statistics of our benchmark in Figure 9.

## A.6 FULL APPLICATION LIST

Here we include the full list of applications involved in our benchmark.

---

**List of Applications**

**Office (30)**: Apple Notes, Apple Reminders, Calendar, Docs, Document Viewer, Evince, Gedit, Google Calendar, Google Docs, Google Keep, Keynote, Lark, Libreoffice, Notability, Note-taking App, Notepad, Notes, Notion, Numbers, Office, Overleaf, Pages, Powerpoint, Spreadsheet, Text Editor, VS Code, WPS Office, Microsoft Word, Xcode, Freeform.

**Media (18)**: Amazon Music, Amazon Prime Video, Iheartradio, Likee, Music, Music Player, Pandora, Pocket FM, Podcast Player, Quicktime, Roku, Sofascore, Spotify, TikTok, Tubi, VLC media player, YouTube, YouTube Music.

**Game (12)**: Arena_of_valor, CS2, Chess, Defense_of_the_ancients_2, Dream, Genshin_impact, Minecraft, Nintendo, Pubg, Red_dead_redemption_2, Steam, The Legend Of Zelda Breath Of The Wild.

**Editing (20)**: 3dviewer, Adobe Acrobat, Adobe After Effects, Adobe Express, Adobe Photoshop, Adobe Photoshop Express, Adobe Premiere Pro, CapCut, Davinci Resolve, Draw.io, Gimp, Paint, PDF Editor, Photo Editing Tool, Photo Editor, Picsart, Procreate, Runway, Snapseed, Video Editing Software.

---

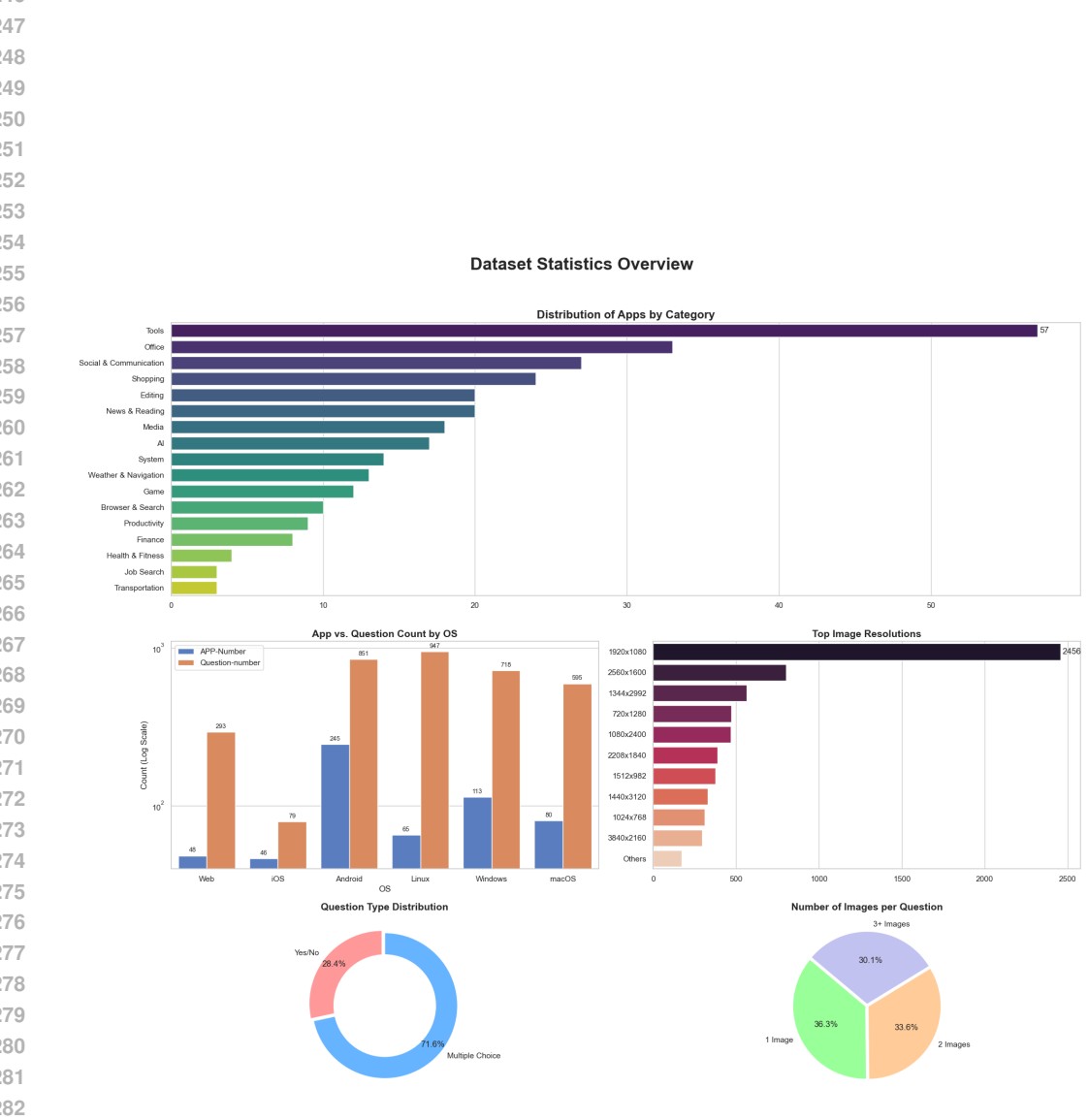

Figure 9: Dataset Statistics Overview

**Social & Communication (28)**: Discord, Facebook, Flickr, Gmail, Google Meet, Google Messages, Imessage, Instagram, LinkedIn, Mail, Messenger, Outlook, Phone, Pinterest, Quora, Reddit, Signal, Slack, Teams Live, Telegram, Threads, Thunderbird, Tumblr, WeChat, Weibo, WhatsApp, X (Twitter), Zoom.

**Shopping (25)**: 12306, Alibaba, Aliexpress, Amazon Shopping, Apartments.com, Applestore, Autoscout24, Autouncle, Booking.com, Car Marketplace, Cars.co.za, Ebay, Edmunds, Expedia, Magento, Offerup, Onestopmarket, Product Listing App, Realtor.com, Redfin, Shop, Taobao, Tripadvisor, Walmart, Wish.

**AI & Tools (17)**: AI Art Generator, Align-anything-dev-omni, Amazon Alexa, Chatbot AI, Chatgpt, Chaton AI, DeepL Translate, Google Translate, Grammarly, Microsoft Copilot, Microsoft Translator, Remix AI Image Creator, Stable Diffusion, Translate, WOMBO Dream, Yandex Translate, Zhiyun Translate.

**Browser & Search (10)**: Bing, DuckDuckGo, Firefox, Google App, Google Chrome, Google Search, Opera, Safari, Web Browser, Web.

**Tools (60)**: Accerciser, Activities, Activity Monitor, App Lock, App Locker, Applock Pro, Automator, Baidu Netdisk, Bluetoothnotificationareaiconwindowclass, Calculator, Camera, Clean, ClevCalc - Calculator, Color Management Utility, Colorsync_utility, Contacts, Control Center, Cursor, Desktop, Dictionary, Digital Color Meter, Disk Utility, Drops, Electron, Email Client, File, File Explorer, File Manager, Files, Filezilla, Finder, Font Book, GPS, Image Viewer, Iphonelockscreen, Kid3, Launcher, Mi Mover, Microsoft Store, Preview, Recorder, Rosetta Stone, Scientific Calculator Plus 991, Script_editor, Search, Shortcuts, Spotlight, Stickies, System Information, System Search, System Settings, Task Manager, Terminal, Totem, ToDesk, Trash, Vim, Voicememos, Vottak, Wallpaper Picker.

**Productivity (9)**: Any.do, Drive, Dropbox Paper, Google Drive, Onedrive, Paperflux, Things, TickTick, Todoist.

**News & Reading (22)**: AP News, BBC News, BBC Sport, Bloomberg, Crimereads, Espn, Forbes, Goodreads, Google News, Google Play Books, Google Scholar, Kindle, Kobo Books, Metacritic, Microsoft News, Newsbreak, Wikidata, Wikipedia, Yahoo Sports, Apple News, Travel Guide App, Travel Review App.

**Weather & Navigation (12)**: Accuweather, Apple Maps, Citymapper, Google Maps, Mapillary, Miuiweather, Msnweather, Navigation App, Openstreetmap, Waze, Weather, Windy.

**Finance (8)**: Alipay, Budgeting App, Investing.com, Paymore, Stocks, Wallet For Your Business, Wallet: Budget Money Manager, Yahoo Finance.

**Health & Fitness (4)**: Fitbit, Fiton, Mideaair, Mifitness.

**Job Search (3)**: Indeed, Job Search By Ziprecruiter, Ziprecruiter.

**Transportation (3)**: Didi, Ryanair, Uber.

**System & Tools (15)**: Android, Android Home Screen, Android Launcher, Android Settings, Android Share Sheet, App Store, Apple, Applibrary, Gnome, Mobile Home Launcher, Mobile Launcher, Mobile Web Browser, OS, Ubuntu, Ubuntu Desktop.

## A.7 ACTION TYPE PREDICTION CONFUSION MATRIX

Figure 10 and Figure 11 show the confusion matrix of tested models on desktop and mobile. All of these models have a tendency for predicting click instead of the right actions.

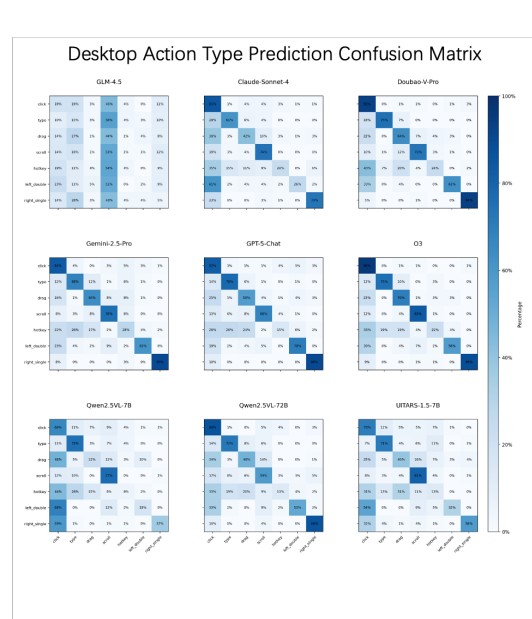

Figure 10: Confusion matrix of action type prediction in desktop.

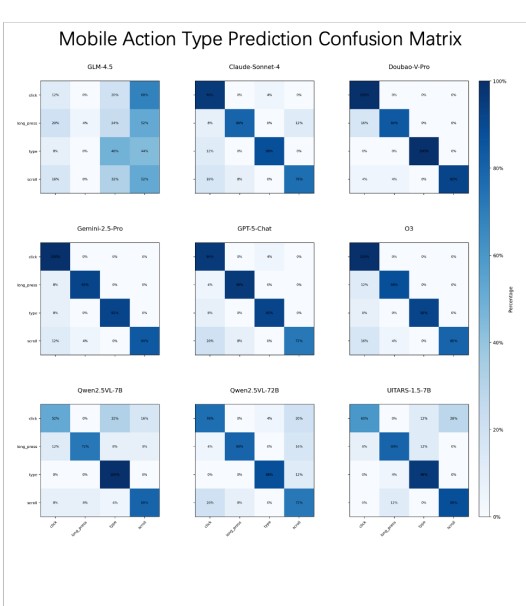

Figure 11: Confusion matrix of action type prediction in mobile.

