# OpenReview forum: "GUI Knowledge Bench: Revealing the Knowledge Gap Behind VLM Failures in GUI Tasks"
_ICLR.cc/2026/Conference — ICLR 2026 Conference Withdrawn Submission_

### Official Review · Reviewer_Qai1 · 2025-10-30

**Soundness:** 3
**Presentation:** 3
**Contribution:** 3
**Rating:** 6
**Confidence:** 4

**Summary:**

This paper investigates why large vision–language models still struggle with automated GUI tasks despite recent advances. The authors hypothesize that missing core GUI knowledge in current VLMs is a key factor behind their performance gap compared to humans. To address this, the paper makes several notable contributions. First, it defines three dimensions of GUI knowledge based on common failure patterns: (1) interface perception, (2) interaction prediction, and (3) instruction understanding. Using these categories, the authors introduce GUI Knowledge Bench, a comprehensive benchmark composed of multiple-choice and yes/no questions derived from over 40,000 GUI screenshots and 400 execution traces across six platforms and 292 applications. This benchmark systematically evaluates what GUI-relevant knowledge is encoded in VLMs prior to any fine-tuning on task-specific data. The paper’s experiments show that while current VLMs can often recognize basic widget functions, they struggle with perceiving system state, predicting interaction outcomes, and verifying task completion. Importantly, the authors demonstrate a close link between performance on the knowledge benchmark and real GUI task success: models with more encoded GUI knowledge perform better on actual GUI automation tasks, and providing missing knowledge (e.g. in the form of operation plans) significantly improves task execution success. Overall, the paper’s contributions include a novel benchmark dataset for GUI knowledge, an analysis revealing specific knowledge gaps in state-of-the-art VLMs, and empirical evidence that addressing these knowledge gaps can improve GUI task automation.

**Strengths:**

1. The paper tackles a crucial and timely problem in multimodal AI - why VLM-driven GUI agents often fail in real scenarios. The authors clearly motivate that beyond reasoning and planning, knowledge of GUI specifics is missing in current models.
2. The introduction of a large-scale benchmark with 3483 knowledge-centric questions across 6 operating systems and 292 applications is a significant contribution.
3. The paper’s breakdown of GUI knowledge into three dimensions – interface perception, interaction prediction, and instruction understanding – is well-grounded in observed failure modes and prior literature.
4. The authors evaluate a wide range of state-of-the-art models, including both closed-source and open-source models. The benchmarking results are detailed per knowledge category and sub-task, allowing for nuanced comparisons. This extensive comparison lends credibility to the findings.
5. The results yield clear, actionable insights.
6. A significant strength is the additional experiment bridging the benchmark to real task execution.

**Weaknesses:**

1. While the paper states that the 3,483 questions were produced via automated generation plus manual annotation, it provides few details on this process. It’s not fully clear how questions and answers were generated or verified for correctness and difficulty.
2. Focus on base VLMs without fine-tuning. The benchmark specifically tests models “prior to downstream training” – i.e., base VLMs without task-specific fine-tuning. This isolates inherent knowledge but also means models are out-of-the-box, not specialized for UI understanding.
3. Evaluation of knowledge integration methods is limited. The paper’s solution implications mainly suggest selecting better base models or augmenting them with knowledge. While it demonstrates that adding operation plans helps one model, it stops short of exploring other ways to inject GUI knowledge (for instance, using the benchmark itself as additional training data or using retrieval during inference). A deeper discussion on concrete strategies (beyond the brief mention of retrieval augmentation) would have been useful to translate the findings into actionable guidance for building improved agents.

**Questions:**

1. How were the 3483 knowledge questions constructed and validated? The paper mentions a mix of automated generation and manual annotation, but clarification is needed on the process. For example, did you use templates to generate questions from execution trajectories, and what steps ensured that the questions have unambiguous correct answers and appropriate difficulty?
2. The OSWorld case study is great evidence of knowledge helping in one setting. Did you observe (or do you anticipate) a strong correlation between a model’s score on GUI Knowledge Bench and its success rate on various downstream GUI tasks (beyond OSWorld)? For example, if Model A scores 10% higher on the benchmark than Model B, is A consistently better when fine-tuned or evaluated on full task benchmarks?

---

> ### Author Response · Authors · 2025-11-21
> **Response to Reviewer4 Qai1**
>
> >**W1.** While the paper states that the 3,483 questions were produced via automated generation plus manual annotation, it provides few details on this process. It’s not fully clear how questions and answers were generated or verified for correctness and difficulty. & **Q1.** How were the 3483 knowledge questions constructed and validated? The paper mentions a mix of automated generation and manual annotation, but clarification is needed on the process. For example, did you use templates to generate questions from execution trajectories, and what steps ensured that the questions have unambiguous correct answers and appropriate difficulty?
>
> **A1.** The construction and validation process is described in each knowledge category’s Task Collection and Curation section (i.e., line 236-242 for interface perception, 281-293 for interaction prediction and 341-349 for goal interpretation.), and the full prompt templates are provided in Appendix A. The detailed generation process involves: a. automatically generate candidate questions from a diverse corpus of 40k+ screenshots in 292 applications, across 6 platforms, ensuring high coverage. b. filter out questions solvable without images to ensure the difficulty of the question. c. manually check the filtered questions to ensure the correctness.
>
> >**W2.** Focus on base VLMs without fine-tuning. The benchmark specifically tests models “prior to downstream training” – i.e., base VLMs without task-specific fine-tuning. This isolates inherent knowledge but also means models are out-of-the-box, not specialized for UI understanding.
>
> **A2.** Our intention is to evaluate off-the-shelf VLMs, which include both:  General-purpose instruction-tuned VLMs (e.g., GPT-5-Chat), and  GUI-specialized models that have undergone additional SFT/RLHF for GUI tasks (e.g., UITARS-1.5-7B).
> Modern general VLMs have encountered vast amounts of GUI data during pre-training. Our results confirm this: models like **o3** achieved highly competitive scores across knowledge categories without specific tuning, validating that "general" models do possess inherent GUI knowledge. Besides, agentic framework such as AgentS2.5 built upon o3 also achieves leading performance on OSworld.
> By comparing general-purpose instruction-tuned VLMs (e.g., GPT-5-Chat), and  GUI-specialized models, we revealed a critical insight. For instance, **UITARS-1.5-7B** (fine-tuned from Qwen2.5-VL) shows clear gains in instruction following and reasoning, but exhibits a noticeable degradation in low-level interface perception compared to its general-purpose counterpart. This suggests that current GUI-specific tuning may enhance high-level planning at the cost of narrowing the model's general perceptual knowledge.

---

> > ### Author Response · Authors · 2025-11-21
> > **Response to Reviewer4 Qai1**
> >
> > >**W3.** Evaluation of knowledge integration methods is limited. The paper’s solution implications mainly suggest selecting better base models or augmenting them with knowledge. While it demonstrates that adding operation plans helps one model, it stops short of exploring other ways to inject GUI knowledge (for instance, using the benchmark itself as additional training data or using retrieval during inference). A deeper discussion on concrete strategies (beyond the brief mention of retrieval augmentation) would have been useful to translate the findings into actionable guidance for building improved agents.
> >
> > **A3.**
> > We believe that possible knowledge injection methods can be categorized as follows:
> >
> > Inference-Time Knowledge Augmentation. To address gaps in operation planning without altering model weights, we propose advancing Retrieval-Augmented Generation (RAG) beyond simple text retrieval. First, given the structured nature of software, future systems can construct GUI Knowledge Graphs from exploration trajectories, utilizing graph traversal algorithms (e.g., BFS) to retrieve optimal action paths or successful sub-goals rather than isolated steps. Second, considering the visual-centric nature of GUIs, effective augmentation must be multimodal. This involves retrieving video-based demonstrations or key-frame sequences that match the current screen state, allowing the agent to mimic complex interaction patterns visually. Finally, dynamic guideline injection is crucial; rather than static instructions, agents should utilize a "selector" mechanism to fetch the most relevant synthesized tutorials or specific operational constraints based on the real-time context.
> >
> > Training-Time Knowledge Internalization. For deficits related to fundamental GUI knowledge, we suggest strategies to embed knowledge directly into the model's parameters. One key direction is continual pre-training on large-scale, unlabeled, or semi-labeled GUI data. This allows models to intrinsically learn general GUI knowledge, layout patterns, and interaction logic (e.g., the function of a "hamburger menu") before tackling specific tasks. Complementing this, we advocate for reasoning-centric instruction tuning. Instead of merely training on (state, action) pairs, future datasets should explicit requires the model to generate the rationale behind an interaction. This forces the model to internalize the causal rules of GUI operations, transforming it from a coordinate predictor into a reasoned decision-maker.
> >
> > Conclusion: Our benchmark serves as the diagnostic foundation to determine which of these paths is necessary. For instance, low scores in the Knowledge dimension suggest a need for the Retrieval strategies mentioned above, while failures in Planning indicate a need for the Reasoning-centric training approaches. We have updated the manuscript to include this structured roadmap.

---

> ### Author Response · Authors · 2025-11-21
> **Response to Reviewer4 Qai1**
>
> >**Q2.** The OSWorld case study is great evidence of knowledge helping in one setting. Did you observe (or do you anticipate) a strong correlation between a model’s score on GUI Knowledge Bench and its success rate on various downstream GUI tasks (beyond OSWorld)? For example, if Model A scores 10% higher on the benchmark than Model B, is A consistently better when fine-tuned or evaluated on full task benchmarks?
>
> **A2.**
>
> To address this, we have added an additional validation study that mirrors the the 'Instruction Understanding' analysis for 'Interface Perception' and 'Interaction Prediction'. This allows us to directly analyse a model's knowledge of a specific element with its ability to operate on that element. Concretely, we transform 39 questions in our benchmark into practical GUI tasks, where the knowledge tested in questions is key to completing the GUI tasks. Two examples are as follows.
> - Knowledge Question: "Are the search results limited to a specific region? (Yes/No/Unknown)"
> Transformed Task: "Set the search results restricted to a specific region: Japan." (Ground Truth: Click the toggle switch).
> - Knowledge Question: "Is the list currently arranged from older to newer items?"  Transformed Task: "Sort the items from oldest to newest according to the time added." (Ground Truth: Click the column header).
> **experimental results**.  We define $S1$ as answering the knowledge question correctly, and $S2$ as successfully executing the corresponding GUI action.
>
> **Table B.**
> | Model | P(S2✓\|S1✓) | P(S2✗\|S1✓) | P(S2✗\|S1✗) | P(S2✓\|S1✗) |
> |-----------------|--------------|--------------|--------------|--------------|
> | claude-sonnet-4               |       20.00% |       80.00% |      100.00% |        0.00% |
> | claude-sonnet-4-5             |        8.33% |       91.67% |      100.00% |        0.00% |
> | doubao-1-5-thinking-vision-pro  |        0.00% |      100.00% |      100.00% |        0.00% |
> | gemini-2.5-pro           |        0.00% |      100.00% |      100.00% |        0.00% |
> | gpt-5-chat                     |        5.56% |       94.44% |      100.00% |        0.00% |
>
> The results establish GUI knowledge as a **necessary but not sufficient condition** for task completion:
>
> Knowledge as Gatekeeper ($P(S2\times|S1\times) \approx 100\%$): Lacking knowledge guarantees execution failure (e.g., 100% for gemini-2.5-pro), validating that knowledge is the strict lower bound for control.
>
> Grounding Bottleneck ($P(S2\checkmark|S1\checkmark)$ is low): Even with correct understanding, execution often fails due to grounding precision.
> Conclusion: this confirms that our benchmark measures a foundational capability. While having knowledge doesn't guarantee success (due to downstream grounding issues), lacking knowledge guarantees failure.
>
> In addition, we reviewed the performance of our tested models on other public GUI-agent benchmarks. Qwen-2.5-VL-72B scores on average higher than Qwen-2.5-VL-7B (63.88% v.s. 45.16%), and in the meantime, Qwen-2.5-VL-72B outperformed qwen-2.5-vl-7b on ScreenSpot-Pro (43.6 v.s. 29.0).(Performance taken from [Qwen2.5-VL](https://arxiv.org/abs/2502.13923),[UI-tars](https://seed-tars.com/1.5/))

---

### Official Review · Reviewer_HyCn · 2025-11-01

**Soundness:** 2
**Presentation:** 1
**Contribution:** 2
**Rating:** 4
**Confidence:** 4

**Summary:**

This paper introduces GUI Knowledge Bench, a diagnostic benchmark that evaluates vision-language models (VLMs) on GUI-specific knowledge across three dimensions: interface perception, interaction prediction, and instruction understanding. The benchmark spans 6 platforms and 292 applications with 3,483 questions, revealing that current VLMs struggle with system state reasoning, action prediction, and task completion verification despite reasonable performance on basic widget recognition.

**Strengths:**

1. Comprehensive Coverage: The benchmark's scope is impressive - covering 6 platforms, 292 applications, and over 40,000 screenshots. This diversity enables robust evaluation across different GUI environments.

2. Interesting Empirical Analysis: The evaluation reveals clear patterns - models perform well on widget function recognition but struggle with system states and interaction dynamics. The confusion matrix showing bias toward "click" actions is insightful.

**Weaknesses:**

1. The benchmark mixes three distinct capabilities: knowledge, perception, and grounding. For example, for some of the problems, it involves clicking on a coordinate which requires strong grounding ability. The mix of various abilities make it harder to understand what causes the deficiency in GUI models.

2. The paper assumes VLMs should possess extensive prior GUI knowledge, but as mentioned by the authors, GUI interfaces are constantly evolving, and have impossible broad coverage. Why should the model possesses knowledge, rather explore and learn by itself in the environment.

3. Poor figure quality and inconsistent fontsizes. Multiple figures (2-5) contain small text.

**Questions:**

1. Setting mimic real agent trial and error: Does using reflection style (multiple iterations, follow [1] settings) improve accuracy in GUI agent trials?

2. Does correlation exist between your GUI knowledge benchmark and other GUI agent benchmarks, and does possessing this knowledge lead to higher accuracy?

[1] Wang, Xingyao, et al. "Mint: Evaluating llms in multi-turn interaction with tools and language feedback."

---

> ### Author Response · Authors · 2025-11-21
> **Response to Reviewer3 HyCn**
>
> >**w1.** the benchmark mixes three distinct capabilities: knowledge, perception, and grounding. For example, for some of the problems, it involves clicking on a coordinate which requires strong grounding ability. The mix of various abilities make it harder to understand what causes the deficiency in GUI models.
>
> **A1.** Our benchmark is specifically designed to decouple these capabilities, ensuring that performance drops can be clearly attributed to knowledge deficits rather than grounding or perception failures.
> - Separating Grounding: Contrary to the reviewer's concern, our benchmark does not require the model to generate precise coordinates. Exact coordinates in options are annotated in the image as shown in Figure 3. Therefore, the grounding demand is minimal.
> - Separating Perception：By explicitly marking the target region using visual prompts, we also minimize the "Visual Search" burden. The relevant region / button is marked with a red bounding box or red point as shown in Figure 2.
>
> >**w2.** The paper assumes VLMs should possess extensive prior GUI knowledge, but as mentioned by the authors, GUI interfaces are constantly evolving, and have impossible broad coverage. Why should the model possesses knowledge, rather explore and learn by itself in the environment.
>
> **A2.** We agree that exploration is vital for handling evolving interfaces. However, we do not restrict the sources of GUI knowledge, but we measure if the GUI models have the sufficient GUI knowledge, regardless of how these knowledge is acquired (from base VLMs or exploration).
> Besides, we believe prior knowledge is the prerequisite for efficient exploration. Just as a computer-literate human learns new software significantly faster than a novice, an agent possessing strong foundational knowledge can navigate and adapt to new environments far more effectively than one starting from scratch.
>
> >**w3.** Poor figure quality and inconsistent fontsizes. Multiple figures (2-5) contain small text.
>
> **A3.** We have improved the figure resolution and unified the font sizes in the revision.
>
> >**Q1.** setting mimic real agent trial and error: Does using reflection style (multiple iterations, follow [1] settings) improve accuracy in GUI agent trials? [1] Wang, Xingyao, et al. "Mint: Evaluating llms in multi-turn interaction with tools and language feedback."
>
> **A1.** Reflection—much like human trial-and-error—can indeed help an agent correct mistakes and acquire task-specific knowledge during interaction, and we agree that it generally improves performance. However, reflection is only effective when the agent possesses sufficient foundational GUI knowledge to begin with.
>
> Without this prior knowledge, reflection often degenerates into random exploration. This not only reduces efficiency, but also raises safety concerns, as the agent may perform irreversible or destructive actions (e.g., deleting files) while “testing” the interface.
>
> For these reasons, our benchmark focuses on evaluating GUI knowledge that a VLM must have. Strong foundational knowledge is what enables reflection mechanisms, such as those in [1], to operate efficiently and safely in real-world GUI environments.
>
>
> >**Q2.** Does correlation exist between your GUI knowledge benchmark and other GUI agent benchmarks, and does possessing this knowledge lead to higher accuracy?
>
> **A2.**
> We compared models from the Claude-Sonnet series and observed that Claude-Sonnet-4.5 (100 steps) outperforms Claude-Sonnet-4 (100 steps) in the OSWorld benchmark (62.9 v.s. 41.4, performance taken from [osworld-leaderboard](https://os-world.github.io)), and in the meantime Claude-Sonnet-4.5 outperformed Claude-Sonnet-4 in our benchmark, suggesting that stronger GUI knowledge contributes to better downstream performance.
>
> In addition, we reviewed the performance of our tested models on other public GUI-agent benchmarks. Qwen-2.5-VL-72B scores on average higher than Qwen-2.5-VL-7B (63.88% v.s. 45.16%), and in the meantime, Qwen-2.5-VL-72B outperformed qwen-2.5-vl-7b on ScreenSpot-Pro (43.6 v.s. 29.0).(Performance taken from [Qwen2.5-VL](https://arxiv.org/abs/2502.13923),[UI-tars](https://seed-tars.com/1.5/))

---

> ### Comment · Reviewer_HyCn · 2025-11-27
> **Thank you for your response**
>
> Thank you for your response. I will keep my ratings.

---

> > ### Author Response · Authors · 2025-11-28
> >
> > Dear Reviewer,
> > Thank you again for your time and consideration.
> > Just to ensure we fully understand your perspective for future improvements, may I kindly ask whether our rebuttal sufficiently addressed your concerns?
> > We would greatly appreciate any brief clarification if there remain key issues we should further refine.
> > Thank you very much.

---

### Official Review · Reviewer_AVXS · 2025-11-01

**Soundness:** 3
**Presentation:** 2
**Contribution:** 3
**Rating:** 6
**Confidence:** 3

**Summary:**

This paper presents GUI Knowledge Bench (GUIKB), a diagnostic benchmark designed to analyze knowledge gaps in Vision-Language Models (VLMs) applied to graphical user interface (GUI) understanding and control. The benchmark categorizes GUI-related reasoning into three dimensions Interface Perception, Interaction Prediction, and Instruction Understanding band builds a large-scale dataset spanning 6 operating systems, 292 real-world applications, and over 3,400 questions derived from around 40,000 screenshots and 400 GUI trajectories.

**Strengths:**

Timely and relevant problem focus It goes beyond measuring overall task success to dissecting the specific types of GUI knowledge involved.

Broad empirical coverage: The benchmark’s diversity across multiple OSes and hundreds of applications is impressive, offering a realistic assessment environment that surpasses previous GUI benchmarks in scope.

Practical community utility: The dataset could serve as a diagnostic suite to evaluate model grounding and reasoning for future GUI or computer-use agents.

**Weaknesses:**

The main conceptual novelty lies in organizing and scaling previous evaluation ideas. It extends earlier benchmarks (e.g., MMBench-GUI, SeeClick, Web-CogBench) in breadth rather than introducing new forms of reasoning or interaction.

The multiple-choice and yes/no question format, combined with visual hints, simplifies grounding and may allow models to exploit linguistic or positional biases instead of demonstrating genuine understanding. Free form questions (going beyond 25% random performance) could be richer.

The OSWorld improvement experiment is intriguing but small in scope and lacks variance or ablation analysis, making the link between benchmark performance and real agent success suggestive rather than proven. The paper seems to lack deeper analysis which can give insights on what current agents lack in terms of decision making.

**Questions:**

Have you tried a free-form response setting (without multiple-choice cues) to confirm that models truly possess GUI knowledge rather than exploiting format bias?

Could you share dataset composition metrics (OS/app balance, interface types, redundancy rate) to verify diversity and coverage?

I am not very confident accepting this paper, decision subject to rebuttal.

---

> ### Author Response · Authors · 2025-11-21
> **Response to Reviewer2 AVXS**
>
> >**W1.** The main conceptual novelty lies in organizing and scaling previous evaluation ideas. It extends earlier benchmarks (e.g., MMBench-GUI, SeeClick, Web-CogBench) in breadth rather than introducing new forms of reasoning or interaction.
>
> **A1.** We clarify that our contribution is not merely extending earlier benchmarks in breadth, nor is our primary goal to introduce new forms of interaction. Instead, our conceptual novelty lies in shifting the evaluation paradigm from "Execution" to "Knowledge."
> Motivation: We focus on evaluating if VLMs have enough GUI knowledge (e.g., interpreting icons, understanding system states) for completing GUI tasks.  Our benchmark is the first to systematically isolate and measure GUI knowledge in VLMs. The differences between our benchmark and MMBench-GUI, SeeClick, Web-CogBench are as follows.
> - MMBench-GUI focuses on general visual perception and coarse functional understanding. Our benchmark targets fine-grained, structured GUI knowledge, with more challenging and more specific knowledge categories.
> - SeeClick centers on grounding, i.e., mapping language to UI elements. In contrast, we intentionally decouple grounding from the evaluation (by providing visual marks on screenshots, as shown in Figure 2, 3), so that we can isolate and measure the knowledge deficits of current VLMs in GUI interactions, which other grounding-based benchmarks cannot reveal.
> - Web-CogBench examines reasoning ability of VLMs on web pages. It does not target OS-level behaviors, application semantics, or general-purpose GUI operation.  Our benchmark covers much broader interfaces (beyond the web) and focuses on evaluating knowledge.
>
> In summary, our contribution is not a broader version of previous benchmarks, but a new evaluation dimension centered on GUI knowledge, motivated by a failure mode that existing benchmarks do not capture. This perspective leads to new insights about VLM capabilities in GUI task automation.
>
>
> >**W2.** The multiple-choice and yes/no question format, combined with visual hints, simplifies grounding and may allow models to exploit linguistic or positional biases instead of demonstrating genuine understanding.
>
> **A2.** We clarify that the choice of multiple-choice questions is not a simplification, but a deliberate design choice to separate the evaluation of GUI knowledge from other capabilities (such as grounding and reasoning).
>
> -Visual Markers: By explicitly marking elements, we intentionally remove the dependency on grounding (as shown in Figure 3, 4.). This ensures that a failure is due to a lack of Knowledge, not an inability to find or click the element.
>
> - Multiple-Choice Format: To verify that our evaluation does not strongly rely on linguistic or positional biases, we performed an additional experiment on 519 multiple-choice questions from the Interface Perception. We randomly permuted all answer options and re-evaluated three representative models. The results are summarized below:
>
> **Table A.**
> | Interface      | Before | After |
> |----------------|-----|--------|
> | Claude-Sonnet-4   |  79.19\% |80.73\%|
> |  GPT-5-Chat| 85.16\% | 85.93\%   |
> |O3 |85.16\% | 85.74\% |
>
> “Before” denotes performance on the original option ordering, and “After” denotes performance after randomly shuffling all answer options. The accuracy remains nearly unchanged across models, indicating that the multiple-choice format does not introduce meaningful positional or linguistic biases. This supports our claim that the benchmark primarily tests GUI knowledge rather than exploiting superficial cues.

---

> ### Author Response · Authors · 2025-11-21
> **Response to Reviewer2 AVXS**
>
> >**W3.** The OSWorld improvement experiment is intriguing but small in scope and lacks variance or ablation analysis, making the link between benchmark performance and real agent success suggestive rather than proven. The paper seems to lack deeper analysis which can give insights on what current agents lack in terms of decision making.
>
> **A3.**
> To address this, we have added an additional validation study that mirrors the the 'Instruction Understanding' analysis for 'Interface Perception' and 'Interaction Prediction'. This allows us to directly analyse a model's knowledge of a specific element with its ability to operate on that element. Concretely, we transform 39 questions in our benchmark into practical GUI tasks, where the knowledge tested in questions is key to completing the GUI tasks. Two examples are as follows.
> - Knowledge Question: "Are the search results limited to a specific region? (Yes/No/Unknown)"
> Transformed Task: "Set the search results restricted to a specific region: Japan." (Ground Truth: Click the toggle switch).
> - Knowledge Question: "Is the list currently arranged from older to newer items?"  Transformed Task: "Sort the items from oldest to newest according to the time added." (Ground Truth: Click the column header).
> **experimental results**.  We define $S1$ as answering the knowledge question correctly, and $S2$ as successfully executing the corresponding GUI action.
>
> **Table B.**
> | Model | P(S2✓\|S1✓) | P(S2✗\|S1✓) | P(S2✗\|S1✗) | P(S2✓\|S1✗) |
> |-----------------|--------------|--------------|--------------|--------------|
> | claude-sonnet-4               |       20.00% |       80.00% |      100.00% |        0.00% |
> | claude-sonnet-4-5             |        8.33% |       91.67% |      100.00% |        0.00% |
> | doubao-1-5-thinking-vision-pro  |        0.00% |      100.00% |      100.00% |        0.00% |
> | gemini-2.5-pro           |        0.00% |      100.00% |      100.00% |        0.00% |
> | gpt-5-chat                     |        5.56% |       94.44% |      100.00% |        0.00% |
>
> The results establish GUI knowledge as a **necessary but not sufficient condition** for task completion:
>
> Knowledge as Gatekeeper ($P(S2\times|S1\times) \approx 100\%$): Lacking knowledge guarantees execution failure (e.g., 100% for gemini-2.5-pro), validating that knowledge is the strict lower bound for control.
>
> Grounding Bottleneck ($P(S2\checkmark|S1\checkmark)$ is low): Even with correct understanding, execution often fails due to grounding precision.
> Conclusion: this confirms that our benchmark measures a foundational capability. While having knowledge doesn't guarantee success (due to downstream grounding issues), lacking knowledge guarantees failure.
>
> From our benchmark and the extra experiments, we observe that current VLM-based GUI agents often fail because they lack critical GUI knowledge needed for decision making. Specifically:
> - Models lack knowledge about how GUI states encode system conditions (e.g., enabled/disabled, selected/unselected, active/inactive). Without this knowledge, agents misinterpret the interface and choose invalid or ineffective actions.
> - Models do not reliably know how actions transform GUI states (e.g., toggling changes a mode; applying a setting triggers a confirmation dialog). This missing interaction knowledge leads to incorrect action sequences and failure to reach the desired final state.
> - Models lack knowledge about what counts as task completion in typical GUI workflows (e.g., changing a setting requires applying it; exporting requires choosing a format). As a result, agents often stop prematurely or pursue incorrect subgoals.

---

> > ### Author Response · Authors · 2025-11-21
> > **Response to Reviewer2 AVXS**
> >
> > >**Q1.** Have you tried a free-form response setting (without multiple-choice cues) to confirm that models truly possess GUI knowledge rather than exploiting format bias?
> >
> > To further confirm that models are not exploiting multiple-choice cues, we additionally evaluate the same 519 questions in a free-form response setting. Models must generate an open-ended textual answer without being shown the candidate options. We then use LLM-as-a-judge to automatically score correctness.
> >
> > Specifically, Claude-Sonnet-4, GPT-5-Chat, and O3 are leveraged to generate free-form answers, and then independently evaluate their correctness. Results are shown below:
> >
> > **Table C.**
> > |judge model  /evaluation model | claude-sonnet-4 | gpt-5-chat | o3|
> > |--|--|--|--|
> > claude-sonnet-4 | 47.21%  |  52.02%  |   **55.11%**
> > gpt-5-chat |    48.55%  |  55.49%   |    **57.23%**
> > o3 |  47.01%  |  53.56%  |  **55.88%**
> >
> > Key findings:
> > - Free-form answering is substantially harder than the multiple-choice setting, i.e., all three models achieve lower scores. This is because models must both recall the correct knowledge and produce the answer in an unconstrained format.
> > - Results are consistent with our testing results: where o3 achieves overall higher ratings.
> >
> >
> >
> > >**Q2.** Could you share dataset composition metrics (OS/app balance, interface types, redundancy rate) to verify diversity and coverage?
> >
> > **A2**
> > To ensure comprehensive coverage and high diversity, we curated the benchmark across 3 major platforms (Web, Mobile, Desktop) and 6 operating systems. The details are presented below:
> >
> > **Table D**
> > | Interface      | Web | Mobile | Desktop |
> > |----------------|-----|--------|---------|
> > | APP-Number     | 48  | 291    | 258     |
> > | Question-number| 293 | 930    | 2260    |
> >
> > **Table E**
> > | os-name        | Web | iOS | Android | Linux | Windows | macOS |
> > |----------------|-----|-----|---------|-------|---------|-------|
> > | APP-Number     | 48  | 46  | 245     | 65    | 113     | 80    |
> > | Question-number| 293 | 79  | 851     | 947   | 718     | 595   |

---

### Official Review · Reviewer_RHPn · 2025-11-01

**Soundness:** 1
**Presentation:** 2
**Contribution:** 2
**Rating:** 2
**Confidence:** 4

**Summary:**

This paper posits that the performance gap between Vision-Language Models (VLMs) and humans in GUI automation stems from a core knowledge deficit, rather than failures in reasoning or planning alone. The authors introduce GUI Knowledge Bench, a diagnostic benchmark designed to evaluate this specific knowledge gap. The benchmark is structured around three dimensions derived from common agent failure modes: Interface Perception (recognizing widgets, layouts, and system states), Interaction Prediction (predicting action types and their effects), and Instruction Understanding (planning and verifying task completion).

**Strengths:**

1. The primary strength is the conceptual shift in evaluation. Instead of focusing on end-to-end task success, which conflates multiple agent capabilities (e.g., reasoning, planning, knowledge, grounding)
2. The benchmark is comprehensive, covering 6 platforms and 292 applications, which ensures the findings are generalizable.

**Weaknesses:**

1. Conflation of "Knowledge" and "Reasoning" in Benchmark: The paper's primary claim is to evaluate GUI "knowledge" (”Different from most existing benchmarks that primarily evaluate task success, which mainly focus on the grounding, reasoning, and planning”) as distinct from "reasoning" and "planning." However, several tasks within the benchmark, particularly in the "Ins Understanding" (e.g., Task Planning) and "Interaction Prediction" dimensions, appear to inherently require complex reasoning or logical capability of LMs.
2. There is a contradiction in the paper's terminology. The authors claim to test 'base VLMs,' but the models listed are clearly instruction-tuned, optimized, and/or RLHF'd (e.g., 'GPT-5-Chat'). 'UITARS-1.5-7B' is itself a fine-tuned GUI agent (sft/rled based on qwen2.5vl). This invalidates the premise of testing 'base' models.
3. The OSWorld validation in Sec 4.4 provides evidence that providing plans improves agent performance. This validates the 'Instruction Understanding' gap. However, the link for 'Interface Perception' and 'Interaction Prediction' is supported primarily by two examples described by words. A stronger validation would require a systematic correlation analysis.

**Questions:**

1. The benchmark's construction relies heavily on 'GPT-5' to generate question-answer pairs. This methodology is concerning as it may introduce artifacts or biases specific to the generator model, which could then be reflected in the evaluation of other VLMs. Does this benchmark test for fundamental GUI knowledge or for knowledge that 'GPT-5' happens to encode well?

---

> ### Author Response · Authors · 2025-11-21
> **Response to Reviewer1 RHPn**
>
> > **W1.** Conflation of "Knowledge" and "Reasoning" in Benchmark: The paper's primary claim is to evaluate GUI "knowledge" (”Different from most existing benchmarks that primarily evaluate task success, which mainly focus on the grounding, reasoning, and planning”) as distinct from "reasoning" and "planning." However, several tasks within the benchmark, particularly in the "Ins Understanding" (e.g., Task Planning) and "Interaction Prediction" dimensions, appear to inherently require complex reasoning or logical capability of LMs.
>
> **A1.** We clarify that this benchmark was designed to minimize the reasoning required so that we can accurately evaluate GUI knowledge in VLMs. We clarify the design choices below.
> 1. Questions in Task Planning are used to evaluate the procedural knowledge, not planning or multi-step reasoning. The questions do not ask the model to construct a plan or generate sub-steps. Instead, the complete operation sequence is already provided in these questions. Thus, this task reduces to verifying whether the order of these operations matches a well-known GUI standard operating procedure. For example, consider the following question from our benchmark: "Task: I am currently working on a ubuntu system but I do not want the notifications to bother me. Can you help me to switch to 'Do not disturb mode'? Please determine whether the following sequence is correct:1. "Toggle the "Do Not Disturb" switch to On."2. "Click anywhere outside the menu to close it. "3. "Click the bell icon in the top bar to open the notifications menu."” Once the underlying procedural knowledge is known—click the bell icon → toggle Do Not Disturb → close the menu—the correctness of the sequence becomes immediately obvious and requires minimal reasoning. This check does not require long-horizon reasoning or generation of novel plans, it only tests whether the model possesses the correct procedural knowledge.
> 2.  Questions in Interaction Prediction examines knowledge of standard GUI interaction conventions. (e.g., buttons are activated by clicking, sliders are adjusted by dragging). These questions probe whether the model understands shared GUI rules—e.g., buttons are activated by clicking, sliders are adjusted by dragging, toggles switch states with a tap, etc. Knowing these rules is the requirement for correctly answering our benchmark questions, without requiring multi-step reasoning.
> 3.  Any remaining reasoning is minimal and already well within the capability range of modern VLMs. The small amount of reasoning involved—such as verifying whether a sequence adheres to a known convention—is extremely basic and not where current models fail. Such reasoning is straightforward and not the source of model failures. Existing VLMs are generally competent at this level of inference. The errors we observe arise not from reasoning deficits but from missing or incorrect GUI knowledge, such as misunderstanding platform-specific procedures, misidentifying interaction patterns.
>
> > **W2.** There is a contradiction in the paper's terminology. The authors claim to test 'base VLMs,' but the models listed are clearly instruction-tuned, optimized, and/or RLHF'd (e.g., 'GPT-5-Chat'). 'UITARS-1.5-7B' is itself a fine-tuned GUI agent (sft/rled based on qwen2.5vl). This invalidates the premise of testing 'base' models.
>
> **A2.** Our intention is to evaluate off-the-shelf VLMs, which include both: General VLMs (e.g., GPT-5-Chat and Qwen2.5-VL), and GUI-specialized models that have undergone additional SFT/RLHF for GUI tasks (e.g., UITARS-1.5-7B).  We aim to characterize how existing, publicly usable models—both general and GUI-tuned—perform under a unified knowledge-centric evaluation framework. Experimental results show that while UITARS-1.5-7B (fine-tuned from Qwen2.5-VL) demonstrates clear gains in instruction understanding, it exhibits a noticeable degradation in low-level interface perception compared to general VLMs. This suggests that current GUI-specific tuning may enhance high-level planning at the cost of narrowing the model's general perceptual knowledge.

---

> ### Author Response · Authors · 2025-11-21
> **Response to Reviewer1 RHPn**
>
> >**W3.** The OSWorld validation in Sec 4.4 provides evidence that providing plans improves agent performance. This validates the 'Instruction Understanding' gap. However, the link for 'Interface Perception' and 'Interaction Prediction' is supported primarily by two examples described by words. A stronger validation would require a systematic correlation analysis.
>
> **A3.** To address this, we have added an additional validation study that mirrors the the 'Instruction Understanding' analysis for 'Interface Perception' and 'Interaction Prediction'. This allows us to directly analyse a model's knowledge of a specific element with its ability to operate on that element. Concretely, we transform 39 questions in our benchmark into practical GUI tasks, where the knowledge tested in questions is key to completing the GUI tasks. Two examples are as follows.
> - Knowledge Question: "Are the search results limited to a specific region? (Yes/No/Unknown)"
> Transformed Task: "Set the search results restricted to a specific region: Japan." (Ground Truth: Click the toggle switch).
> - Knowledge Question: "Is the list currently arranged from older to newer items?"  Transformed Task: "Sort the items from oldest to newest according to the time added." (Ground Truth: Click the column header).
>
> **experimental results**.  We define $S1$ as answering the knowledge question correctly, and $S2$ as successfully executing the corresponding GUI action.
>
>
> **Table A.**
>
> | Model | P(S2✓\|S1✓) | P(S2✗\|S1✓) | P(S2✗\|S1✗) | P(S2✓\|S1✗) |
> |-----------------|--------------|--------------|--------------|--------------|
> | claude-sonnet-4               |       20.00% |       80.00% |      100.00% |        0.00% |
> | claude-sonnet-4-5             |        8.33% |       91.67% |      100.00% |        0.00% |
> | doubao-1-5-thinking-vision-pro  |        0.00% |      100.00% |      100.00% |        0.00% |
> | gemini-2.5-pro           |        0.00% |      100.00% |      100.00% |        0.00% |
> | gpt-5-chat                     |        5.56% |       94.44% |      100.00% |        0.00% |
>
> The results establish GUI knowledge as a **necessary but not sufficient condition** for task completion:
>
> Knowledge as Gatekeeper ($P(S2\times|S1\times) \approx 100\%$): Lacking knowledge guarantees execution failure (e.g., 100% for gemini-2.5-pro), validating that knowledge is the strict lower bound for control.
>
> Grounding Bottleneck ($P(S2\checkmark|S1\checkmark)$ is low): Even with correct understanding, execution often fails due to grounding precision.
> Conclusion: this confirms that our benchmark measures a foundational capability. While having knowledge doesn't guarantee success (due to downstream grounding issues), lacking knowledge guarantees failure.
>
> >**Q1.** The benchmark's construction relies heavily on 'GPT-5' to generate question-answer pairs. This methodology is concerning as it may introduce artifacts or biases specific to the generator model, which could then be reflected in the evaluation of other VLMs. Does this benchmark test for fundamental GUI knowledge or for knowledge that 'GPT-5' happens to encode well?
>
> **A1.** In this work, we implemented a rigorous, multi-stage quality control pipeline to mitigate the generation bias problem, ensuring the benchmark evaluates fundamental GUI knowledge rather than model-specific artifacts.
> First, instead of unconstrained generation, we collect example questions based on common GUI agent failure modes and build different prompts for each categories, ensuring the questions targets real-world GUI task related deficiencies. These examples and prompts are shown in Appendix A.
> Second, we applied an filtering step using Qwen-2.5-VL-7B to eliminate QA pairs that can be directly answered based on the textual questions (e.g., obvious question-answer pairs like 'How to submit the form? click submit button to submit'), ensuring the quality of the generated data.
> Finally, we performed manual annotation for all corresponding regions and checked the correctness of these questions.

---

### Author Response · Authors · 2025-12-03
**General Response**

we summarize our work (GUI Knowledge Bench) and the major updates during the rebuttal as following:

**1. Reviewer Consensus:** Reviewers unanimously recognize the value of our work and the necessary paradigm shift toward knowledge-based evaluation.
*    **Novel Evaluation Paradigm:** The systematic attempt to isolate GUI knowledge from planning, reasoning, and grounding is a key contribution to the field.
*    **Broad Coverage:** The benchmark's curated diversity across multiple platforms and OSs is critical for realistic evaluation
*    **Relevance to Real-World Tasks:** The benchmark establishes a direct relationship between a model's foundational GUI knowledge and its ultimate success or failure on real-world GUI tasks.


**2. Key Rebuttal Updates**
*    **Validity of Knowledge Evaluation: Decoupling Knowledge from Grounding & Reasoning** (Addressing Reviewer RHPn, AVXS, HyCn): We clarify our design principles for isolating knowledge: By explicitly marking target regions, we remove the burden of visual grounding; by utilizing a multiple-choice format, we exclude the difficulty of open-ended text generation; and by focusing on atomic, factual questions, we minimize the need for complex reasoning. **These design choices ensure that the bottleneck for answering is strictly the model's GUI knowledge.**
*    **Correlation with Downstream Tasks: Knowledge as a "Gatekeeper"** (Addressing Reviewer RHPn, AVXS, HyCn, Qai1)：We add a new validation experiment transforming knowledge questions into executable tasks (for Interface Perception/Interaction Prediction knowledge). Results show that when a model answers a knowledge question incorrectly, its probability of failing the corresponding task is $\approx 100\%$, **establishing a close link between our benchmark scores and real-world task failure.** Besides, in the original manuscript, we successfully improve the agent success rate by injecting plan knowledge (for Instruction Understanding Knowledge). These experiments validate the importance of our knowledge bench.
*    **Rationality of Dataset construction & Evaluation paradigm** (Addressing Reviewer RHPn, AVXS, Qai1): We add clarification on our benchmark's multi-stage construction pipeline, including the following steps: collecting diverse screenshots, generating data through prompts, automatically filtering with Qwen-2.5-VL, manually annotating relevant regions, and conducting thorough human verification, which bring the reliability of our data and alleviates concerns regarding potential artifacts or biases from the generator model (GPT-5). Building on this, we adopt a multiple-choice evaluation format to provide a clearer and more reliable assessment, avoiding the instability commonly seen in LLM-as-judge scoring for free-form answers. To confirm the robustness of this evaluation design, we further perform randomized option permutation and free-form answer tests. The consistent model rankings across these settings indicate that **our benchmark measures genuine GUI knowledge rather than relying on linguistic or positional biases**.

---

### Note · Authors · 2026-01-27

I have read and agree with the venue's withdrawal policy on behalf of myself and my co-authors.

---

### Meta-Review · Area_Chair_P8sQ · 2026-01-01

**Summary:**

This submission argues that the human–VLM gap in GUI task automation is largely driven by missing “core GUI knowledge,” and introduces GUI Knowledge Bench to measure that knowledge across three dimensions (Interface Perception, Interaction Prediction, Instruction Understanding) using MCQ/Yes–No questions with marked target regions. Reviewer sentiment is mixed (one clear reject, two marginal-above-threshold but “would not mind if rejected,” and one marginal-below-threshold but “would not mind if accepted”), suggesting a borderline case.

However, the decision-critical concerns are about construct validity (whether the benchmark truly isolates “knowledge” from reasoning/planning), dataset reliability and generator bias (heavy LLM-generated QA), terminology/claim alignment (“base VLMs” vs instruction-tuned / GUI-specialized models), and insufficiently strong evidence that benchmark scores systematically predict real-world agent performance. While the rebuttal addresses some points, key issues remain outstanding, motivating a Reject recommendation.

**Reviewer Concerns:**

### Concerns addressed (fully or partially) by the rebuttal

- MCQ/Yes–No format may introduce positional/linguistic bias (AVXS): The rebuttal reports an option-permutation test with largely unchanged accuracies/rankings, and additionally provides a free-form setting comparison. This helps mitigate the “format bias” concern, though the free-form evaluation introduces its own limitations (e.g., dependence on judge models).

- Insufficient transparency on dataset composition and construction details (AVXS, Qai1): The rebuttal adds composition statistics (platform/OS/app/question counts) and clarifies a multi-stage pipeline (generation, automated filtering, region annotation, human verification). This improves clarity, but the paper still does not report some standard dataset-quality diagnostics (e.g., ambiguity rate / disagreement statistics if multiple annotators were involved, or systematic audits of generator artifacts), which would further strengthen the reliability claim.

- Weak linkage between knowledge scores and downstream task success (RHPn, AVXS, Qai1, HyCn): The rebuttal adds a small validation by converting 39 knowledge questions into executable tasks and reporting conditional probabilities (knowledge correctness vs task success/failure). This is helpful directionally, but remains limited in scale and interpretability (knowledge appears necessary but far from sufficient).

### Key concerns still outstanding

- Construct validity: “Knowledge” vs “Reasoning/Planning” not convincingly separated (RHPn, HyCn): The rebuttal argues remaining reasoning is “minimal,” but this is largely asserted rather than demonstrated via rigorous construct definitions, task redesign, or ablations. Several sub-tasks—especially within Instruction Understanding and parts of Interaction Prediction—can plausibly require nontrivial procedural verification or inference beyond pure recall. The central claim of “isolating knowledge” is therefore not fully substantiated.

- Claim framing vs model set mismatch (“base VLMs”) (RHPn): The rebuttal reframes the goal as evaluating “off-the-shelf” models (including instruction-tuned and GUI-specialized). This helps explain the experimental choices, but it weakens the original narrative about measuring “base” model prior knowledge and complicates the interpretation of conclusions “prior to downstream training.” The paper’s terminology and the scope of its claims remain misaligned.

- Generator-model artifacts/bias remain a major risk (RHPn): Even with filtering and human checks described, the benchmark is still heavily generated using a powerful LLM, raising the possibility of systematic artifacts (templates, phrasing patterns, option construction biases) that could affect cross-model comparisons. The rebuttal does not provide sufficiently strong auditing evidence to establish that the benchmark measures fundamental GUI knowledge rather than generator-specific regularities.

- Insufficiently strong evidence of predictiveness for real agent success (AVXS, RHPn, HyCn, Qai1): The additional 39-question task conversion is small and yields very low success rates even when the knowledge question is correct, highlighting that knowledge is not sufficient and that other bottlenecks dominate. Given the limited sample size and the “knowledge-as-gatekeeper” framing (necessary but not sufficient), the paper’s implication that the benchmark can reliably select models “with greater potential” for downstream training still appears stronger than what is directly evidenced.

**Reviewer Scores:**

Reviewer RHPn: 2 (Reject, not good enough) . Core concerns (construct separation, “base” terminology contradiction, generator bias) are not convincingly resolved; the added correlation evidence is limited.

Reviewer AVXS: 6 (Marginally above acceptance threshold; ok with reject) . Rebuttal directly addresses MCQ bias and dataset composition; still, concerns about novelty and strength of downstream linkage may persist.

Reviewer HyCn: 4 (Marginally below threshold; ok with accept). The reviewer’s conceptual skepticism (mixed capabilities; whether prior knowledge is required vs exploration) is not fully resolved by rebuttal arguments.

Reviewer Qai1: 6 (Marginally above threshold; ok with reject). Additional construction details and validation help, but broader predictiveness/impact claims remain only partially supported.

---

### Decision · Program_Chairs · 2026-01-26

Reject